# Towards Scalable Oversight via Partitioned Human Supervision

**Ren Yin**[1,2] **Takashi Ishida**[2,1] **Masashi Sugiyama**[2,1]
[1]The University of Tokyo  [2]RIKEN

## Abstract

As artificial intelligence (AI) systems approach and surpass expert human performance across a broad range of tasks, obtaining high-quality human supervision for evaluation and training becomes increasingly challenging. Our focus is on tasks that require deep knowledge and skills of multiple domains, where this bottleneck is severe. Unfortunately, even the best human experts are knowledgeable only in a single narrow area, and will not be able to evaluate the correctness of advanced AI systems on such superhuman tasks. However, based on their narrow expertise, humans may provide a weak signal, i.e., a *complementary label* indicating an option that is incorrect. For example, a cardiologist could state that "this is not related to any cardiovascular disease," even if they cannot identify the true disease. Based on this weak signal, we propose a scalable oversight framework that enables us to evaluate frontier AI systems without the need to prepare the ground truth. We derive an *unbiased* estimator of top-1 accuracy from complementary labels and quantify how many complementary labels are needed to match the variance of ordinary labels. We further introduce two estimators to combine scarce ordinary labels with abundant complementary labels. We provide finite-sample deviation guarantees for both complementary-only and the mixed estimators. Empirically, we show that we can evaluate the output of large language models without the ground truth, if we have complementary labels. We further show that we can train an AI system with such weak signals: we show how we can design an agentic AI system automatically that can improve itself with this partitioned human supervision. Our code is available at ○ Towards Scalable Oversight via Partitioned Human Supervision.

## 1 Introduction

As foundation models and artificial intelligence (AI) systems (OpenAI, 2025a; Anthropic, 2025; DeepSeek, 2025; DeepMind, 2025) approach and in some areas surpass expert human performance, supervision itself becomes a key bottleneck. Current alignment pipelines such as supervised fine-tuning (SFT), reinforcement learning from human feedback (RLHF; Ziegler et al. (2019); Stiennon et al. (2020); Ouyang et al. (2022)), or reinforcement learning from verifiable rewards (RLVR; Shao et al. (2024); Yu et al. (2025); Lambert et al. (2025)) presuppose that humans can reliably provide supervision or design verifiers for training and evaluation. Yet for the superalignment regime (Bowman et al., 2022), future models will tackle problems whose solutions are too technical or too cross-disciplinary for any single human to verify comprehensively. When we cannot produce ground truth or prepare automated verifiers, how should we *evaluate* and *train* such systems?

We begin from an observation about the expertise in high-skill domains in the human society (Wuchty et al., 2007). As tasks grow in difficulty, human experts specialize ever more narrowly: cardiology vs. oncology in medicine, sector-specific analysts in finance, or sub-subfields in science. Specialists can often make *local* determinations with high precision, e.g., a cardiologist might say, "this is *not* a cardiovascular disease," while an oncologist might point out, "this pattern is never seen in the field of oncology." While such judgments may fail to positively certify correctness, they readily certify that certain options are wrong. In other words, even when positive ground truth is out of reach, human experts can often provide reliable *complementary labels* (Ishida et al., 2017; Yu et al., 2018; Wang et al., 2024a; 2025): labels indicating one or more options that are definitely the incorrect.

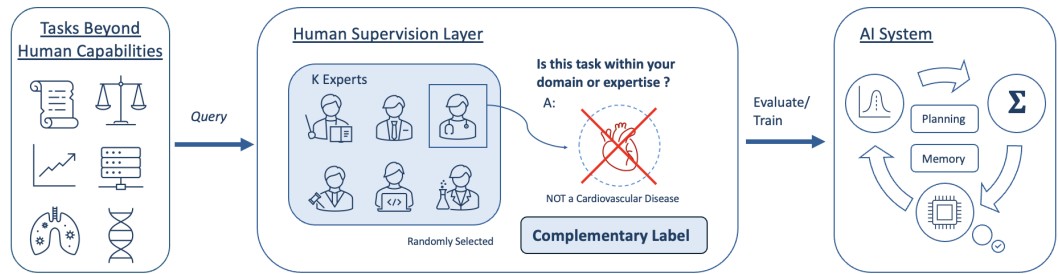

Figure 1: Superhuman tasks are routed to a human expert who can only give a weak signal that they are not capable of solving the task. We show that this weak signal, i.e., *complementary label*, can be used to evaluate/train an AI system.

This paper develops a scalable oversight (Amodei et al., 2016) framework that treats such *partitioned human supervision* as a first-class supervision primitive. Our starting point is a multi-choice evaluation setting where ordinary (true) labels may be scarce or unavailable, but complementary labels arise naturally from partitioned expertise and are abundant. We show that one can utilize these complementary labels to propose an unbiased estimator of top-1 accuracy for an AI system under test, and combine them with scarce ordinary labels for refinement of the estimator. The estimators allow us to (i) *evaluate* frontier models without preparing any ground truth, and (ii) use these evaluations as learning signals to *design* agentic systems on tasks beyond expert human abilities.

Our work complements many other active lines of research on scalable oversight. For example, *weak-to-strong generalization* (Burns et al., 2024) studies whether strong models trained on weak (noisy) supervisors can exceed those supervisors, possibly by eliciting knowledge that the strong model already possesses. *Easy-to-hard generalization* (Sun et al., 2024) trains evaluators on easier problems and applies them to supervise harder ones, by leveraging the assumption that evaluation is easier than generation. *Debate* (Khan et al., 2024) elicits truth via adversarial arguments that a judge selects among. *Reinforcement learning from AI feedback* or *Constitutional AI* (Bai et al., 2022) replaces human preferences with AI-generated feedback. *Recursive task decomposition* (Wu et al., 2021) breaks the problem into leaf subtasks that humans can directly label or compare, for tasks that are easily decomposable. We address a new, orthogonal axis: when human experts cannot supply correct answers for hard, cross-domain instances, can we use their *partitioned* "this option is wrong" judgments to obtain performance estimates and useful training signals?

We formalize a labeling protocol in which each instance is routed to a randomly chosen domain specialist responsible for exactly one option (class). The specialist either confirms that option (yielding an ordinary label) or confidently rejects it (yielding a complementary label). Under this uniform wrong-index design, we derive an *unbiased* estimator of accuracy from complementary labels alone via a simple linear correction, and characterize its variance. We then introduce two *mixture estimators* that combine ordinary and complementary labels: (i) an inverse-variance–weighted (IVW) estimator with a practical plug-in form, and (ii) a closed-form maximum-likelihood estimator (ML).

We empirically validate the framework in three settings. First, we perform *statistical validation* on many popular large language model (LLM) benchmarks, which confirms that we do not need the ground truth to evaluate the performance of AI systems experimentally, such as LLMs. We focus on *real-world evaluation* on tasks that cannot be solved by a single expert human with only narrow expertise. As a proof-of-concept, we use finance and medical benchmarks to demonstrate that partitioned feedback from sector analysts or highly-specialized doctors enable accurate model evaluation without the ability to solve the task alone. Finally, we show that this weak signal can be used as a training signal for an AI system. More specifically, we use complementary labels for *agentic training*: we replace ordinary accuracy with our estimator as the fitness signal inside agent search pipelines, and show that this improves downstream performance, demonstrating a pathway to training when only complementary feedback is available.

Our main contributions are as follows: 1) We introduce scalable oversight via partitioned human supervision, a protocol that exploits real-world specialization to collect *complementary labels* at scale for superhuman tasks. 2) We derive an unbiased complementary-label estimator for top-1 accuracy, analyze its variance, and show how many complementary labels are needed to match the variance of the estimator obtained from ordinary labels. We further propose two mixture estimators together

with finite-sample bounds and practical plug-in implementations. 3) We demonstrate empirically that these estimators enable both evaluation and training without the full ground truth.

## 2 METHODOLOGY

In this section, we explain our problem setup and notations and explain our proposed method.

**Setup and Notations** We consider a multiple-choice evaluation task with $K$ options per item (e.g., $K = 4$ for choices A, B, C, and D). Formally, each item is represented by an input $x \in \mathcal{X}$, where $\mathcal{X}$ denotes the input domain (e.g., the space of question texts). Each item has a correct label $y \in \{1, \ldots, K\}$, representing the index of the correct option after any shuffling. We call this the ordinary label. Let $(X, Y)$ denote a random pair drawn from an identical joint distribution $\mathcal{D}$ over $\mathcal{X} \times \{1, \ldots, K\}$.

Equivalently, we can regard to first sample item $X$, and then a human expert provides label $Y$ based on the underlying conditional distribution $p(Y|X)$. An AI system (e.g., a machine learning classifier, a single LLM, or an LLM agent/workflow) produces a prediction $\hat{Y} = f(X) \in \{1, \ldots, K\}$. For an item with an ordinary label $y$, define the indicator $Z = \mathbb{I}\{\hat{Y} = Y\} \in \{0, 1\}$. The (population) top-1 accuracy is then $A = \Pr(Z = 1) = \Pr(f(X) = Y)$, where $\Pr(\cdot)$ denotes probability with respect to the joint distribution $\mathcal{D}$.

We assume the complementary label is drawn uniformly at random among the non-$y$ classes (Ishida et al., 2017):

$$p(\bar{Y} = k \mid Y) = \frac{1}{K - 1}, \quad \forall k \neq Y. \tag{1}$$

That is, for each item, the revealed wrong index is sampled uniformly from the $K-1$ non-$Y$ classes. For an item with a complementary label $\bar{Y} \neq Y$, define $W = \mathbb{I}\{\hat{Y} \neq \bar{Y}\} \in \{0, 1\}$. The assumption in Eq. 1 is enforced by the following data collection protocol. We assume that we have a set of questions $\{x_i\}_{i=1}^N$ and $K$ annotators, where each annotator is a human expert in the field of one of the classes. For each question $x_i$, choose an annotator randomly (let's say an annotator in charge of the $k$-th class is chosen), and ask "Is the answer of question $x_i$ class $k$?". If the answer is yes, we save this in the ordinary-label set as an item with (ordinary) class $k$ label. If the answer is no, we save this in the complementary-label set as an item with (complementary) class $k$ label. Under this protocol, the uniform sampling assumption in Eq. 1 holds by construction (Appendix 1). Since each expert is queried with equal probability and only performs a binary compatibility check for their designated class, the potential for systematic annotation bias is substantially reduced.

Finally, we will have two disjoint evaluation sets, drawn from the same distribution and evaluated by the same system: 1) An *ordinary-label set* of size $n_\mathrm{o}$, yielding realizations $z_1, \ldots, z_{n_\mathrm{o}}$ and total correct count $S_\mathrm{o} = \sum_{j=1}^{n_\mathrm{o}} z_j$. 2) *complementary-label set* of size $n_\mathrm{c}$, yielding realizations $w_1, \ldots, w_{n_\mathrm{c}}$ and total "weakly-correct"[1] count $S_\mathrm{c} = \sum_{i=1}^{n_\mathrm{c}} w_i$. The total number of samples is $N = n_\mathrm{o} + n_\mathrm{c}$.

### 2.1 ESTIMATOR FROM COMPLEMENTARY LABELS

It is straightforward to estimate the system accuracy $A$ by using ordinary labels as $\widehat{A}_\mathrm{ord} = \frac{S_\mathrm{o}}{n_\mathrm{o}}$. If we want to use complementary labels to estimate $A$, a naive approach would be to count the proportion of "avoiding the complementary label," i.e., $\hat{q} = \frac{1}{n_\mathrm{c}} \sum_{i=1}^{n_\mathrm{c}} w_i$. However, $\hat{q}$ is not an unbiased estimator of the accuracy. Thus, we propose the following complementary-label estimator:

**Corollary 1.** *Under the assumption in Eq. (1), the estimator*

$$\widehat{A}_\mathrm{comp} = (K - 1)\hat{q} - (K - 2) \tag{2}$$

*is unbiased for $A$, where $K \geq 3$ is the number of choices.*

We show the proof in Appendix A. This linear correction is precisely the 0–1 loss specialization of the risk-rewrite identity of Eqs. (8) and (10) in Ishida et al. (2019); replacing expectations by sample means yields our unbiased estimator. A full derivation is provided in Appendix B.

---

[1] Here "weakly-correct" means that the model prediction is *not* equal to the complementary label. For example, when $K = 4$, avoiding the complementary label still leaves a $\frac{2}{3}$ chance of being wrong.

Even if we have an unbiased estimator with only complementary labels, the variance is expected to become larger compared with the case when we use the same number of ordinary labels. This is because complementary labels are "weaker" than ordinary labels in the sense that $K - 1$ different complementary labels is equivalent to 1 ordinary label. We first confirm this as follows. Since $W$ is Bernoulli with mean $q = \frac{A+K-2}{K-1}$, we obtain

$$\mathrm{Var}(\widehat{A}_{\mathrm{comp}}) = (K-1)^2 \, \mathrm{Var}\left( \tfrac{1}{n_{\mathrm{c}}} \sum_{i=1}^{n_{\mathrm{c}}} W_i \right) = \frac{(K-1)^2}{n_{\mathrm{c}}} \, q(1-q) = \frac{(A+K-2)(1-A)}{n_{\mathrm{c}}}. \quad (3)$$

This expression depends on the unknown $A$. In practice, a plug-in estimator is used. For the ordinary-label dataset, the number of correct predictions satisfies $S_{\mathrm{o}} \sim \mathrm{Bin}(n_{\mathrm{o}}, A)$, where $\mathrm{Bin}(n, p)$ denotes the binomial distribution with parameters $n$ (number of independent trials) and $p$ (success probability in each trial). Thus, $\mathbb{E}[S_{\mathrm{o}}/n_{\mathrm{o}}] = A$, and $\mathrm{Var}(S_{\mathrm{o}}/n_{\mathrm{o}}) = A(1-A)/n_{\mathrm{o}}$, where $\mathbb{E}[\cdot]$ and $\mathrm{Var}(\cdot)$ denote expectation and variance, respectively, taken with respect to the underlying population distribution $\mathcal{D}$ over $(x, y)$. Comparing this with Eq. (3), the variance ratio is

$$\frac{\mathrm{Var}(\widehat{A}_{\mathrm{comp}})}{\mathrm{Var}(S_{\mathrm{o}}/n_{\mathrm{o}})} = \frac{(A+K-2)\, n_{\mathrm{o}}}{A \, n_{\mathrm{c}}}. \quad (4)$$

Hence, when $n_{\mathrm{c}} = n_{\mathrm{o}}$, the complementary estimator has larger variance for all $K \geq 3$, as expected. Next, it would be interesting to derive how much more complementary labels we will need to prepare to have the same amount of variance as the ordinary label case. To match the variance of the ordinary estimator, the size of the complementary dataset must be

$$n_{\mathrm{c}} = \left( 1 + \frac{K-2}{A} \right) n_{\mathrm{o}}. \quad (5)$$

That is, the larger $A$ is, the fewer additional complementary labels are needed. Since $A$ is unknown, one can use a small ordinary-labeled set to bound $A$ and infer the required $n_{\mathrm{c}}$, motivating the combined estimator in § 2.2.

## 2.2 Combining Ordinary and Complementary Labels

We now combine information from both datasets. Recall $\widehat{A}_{\mathrm{ord}} = \frac{S_{\mathrm{o}}}{n_{\mathrm{o}}}$ and $\widehat{A}_{\mathrm{comp}} = (K-1)\frac{S_{\mathrm{c}}}{n_{\mathrm{c}}} - (K-2)$ are both unbiased under the independent and identically distributed (i.i.d.) sampling assumption.

**Inverse Variance Weighted Estimator (IVW)** Consider the linear combination $\widehat{A}_{\mathrm{mix}} = w \, \widehat{A}_{\mathrm{ord}} + (1-w) \, \widehat{A}_{\mathrm{comp}}$ where $w \in [0, 1]$. Because the two components are independent and unbiased (disjoint samples), $\widehat{A}_{\mathrm{mix}}$ is unbiased with variance

$$\mathrm{Var}(\widehat{A}_{\mathrm{mix}}) = w^2 \frac{A(1-A)}{n_{\mathrm{o}}} + (1-w)^2 \frac{(K-1)^2}{n_{\mathrm{c}}} q(1-q), \quad q = \frac{A+K-2}{K-1}.$$

Minimizing it w.r.t. $w$ yields

$$w^\star = \frac{\frac{(K-1)^2}{n_{\mathrm{c}}} q(1-q)}{\frac{A(1-A)}{n_{\mathrm{o}}} + \frac{(K-1)^2}{n_{\mathrm{c}}} q(1-q)}. \quad (6)$$

The corresponding minimum variance equals $\mathrm{Var}_{\min}(\widehat{A}_{\mathrm{mix}}) = \frac{\frac{A(1-A)}{n_{\mathrm{o}}} \cdot \frac{(K-1)^2}{n_{\mathrm{c}}} q(1-q)}{\frac{A(1-A)}{n_{\mathrm{o}}} + \frac{(K-1)^2}{n_{\mathrm{c}}} q(1-q)}$. Since $A$ is

unknown, we use plug-in estimators: $\widehat{\mathrm{Var}}(\widehat{A}_{\mathrm{ord}}) = \frac{\widehat{A}_{\mathrm{ord}}(1-\widehat{A}_{\mathrm{ord}})}{n_{\mathrm{o}}}$, and $\widehat{\mathrm{Var}}(\widehat{A}_{\mathrm{comp}}) = \frac{(K-1)^2}{n_{\mathrm{c}}} \hat{q}(1-\hat{q})$, $\hat{q} = \frac{S_{\mathrm{c}}}{n_{\mathrm{c}}}$. Then

$$\widehat{w} = \frac{\widehat{\mathrm{Var}}(\widehat{A}_{\mathrm{comp}})}{\widehat{\mathrm{Var}}(\widehat{A}_{\mathrm{ord}}) + \widehat{\mathrm{Var}}(\widehat{A}_{\mathrm{comp}})}, \text{ where } \widehat{A}_{\mathrm{IVW}} = \widehat{w} \, \widehat{A}_{\mathrm{ord}} + (1-\widehat{w}) \, \widehat{A}_{\mathrm{comp}}. \quad (7)$$

Its (plug-in) variance is the harmonic mean:

$\widehat{\mathrm{Var}}(\widehat{A}_{\mathrm{IVW}}) = \widehat{\mathrm{Var}}(\widehat{A}_{\mathrm{ord}})\,\widehat{\mathrm{Var}}(\widehat{A}_{\mathrm{comp}})/(\widehat{\mathrm{Var}}(\widehat{A}_{\mathrm{ord}}) + \widehat{\mathrm{Var}}(\widehat{A}_{\mathrm{comp}}))$. Substituting $q = \frac{A+K-2}{K-1}$ into Eq. (6) gives

$$w^{\star} = \frac{(A+K-2)\,n_{\mathrm{o}}}{A\,n_{\mathrm{c}} + (A+K-2)\,n_{\mathrm{o}}}, \tag{8}$$

and replacing $A$ by a pilot $\tilde{A}$ (e.g. $\tilde{A} = \widehat{A}_{\mathrm{ord}}$), yields a closed-form estimator of $w^{*}$. If we prepare a separate set other than $n_{\mathrm{c}} + n_{\mathrm{o}}$ samples to estimate the weight, we can maintain the unbiasedness of $\hat{A}_{\mathrm{mix}}$; if not (as in our experiments), we will show in § 2.3 that we have a consistent estimator.

**Closed-Form Maximum-Likelihood Estimator (ML)**  The ordinary set induces a likelihood term in $A$ given by $S_{\mathrm{o}} \sim \mathrm{Bin}(n_{\mathrm{o}}, A)$. The complementary set induces a likelihood term $S_{\mathrm{c}} \sim \mathrm{Bin}(n_{\mathrm{c}}, q)$ with $q = \frac{A+K-2}{K-1}$. Assuming independence between the two datasets, the joint log-likelihood is

$$\ell(A) = S_{\mathrm{o}} \log A + (n_{\mathrm{o}} - S_{\mathrm{o}}) \log(1-A) + S_{\mathrm{c}} \log q + (n_{\mathrm{c}} - S_{\mathrm{c}}) \log(1-q).$$

Setting $\partial\ell/\partial A = 0$ yields the score equation $\frac{S_{\mathrm{o}}}{A} - \frac{n_{\mathrm{o}} - S_{\mathrm{o}}}{1-A} + \frac{S_{\mathrm{c}}}{A+K-2} - \frac{n_{\mathrm{c}} - S_{\mathrm{c}}}{1-A} = 0$, which simplifies to a quadratic form in $A$. Let $T_{\mathrm{o}} = n_{\mathrm{o}} - S_{\mathrm{o}}$, $T_{\mathrm{c}} = n_{\mathrm{c}} - S_{\mathrm{c}}$, $\alpha = N$, $\beta = (K-2)(T_{\mathrm{o}} + T_{\mathrm{c}}) + (K-3)S_{\mathrm{o}} - S_{\mathrm{c}}$, $\gamma = -(K-2)S_{\mathrm{o}}$. Then the ML is the unique root in $[0,1]$:

$$\widehat{A}_{\mathrm{ML}} = \frac{-\beta + \sqrt{\beta^2 - 4\alpha\gamma}}{2\alpha}. \tag{9}$$

A large-sample standard error is $\mathrm{se}(\widehat{A}_{\mathrm{ML}}) \approx \left[ \frac{n_{\mathrm{o}}}{\widehat{A}_{\mathrm{ML}}(1-\widehat{A}_{\mathrm{ML}})} + \frac{n_{\mathrm{c}}}{(K-1)^2\,\hat{q}(1-\hat{q})} \right]^{-1/2}$, based on observed or expected information.

If $n_{\mathrm{c}} = 0$, Eq. (9) reduces to $\widehat{A}_{\mathrm{ord}}$. If $n_{\mathrm{o}} = 0$, it reduces to $\widehat{A}_{\mathrm{comp}}$. For $K = 4$, one always has $\hat{q} \in [\frac{2}{3}, 1]$ when $A \in [0,1]$, which bounds the complementary-component variance. Moreover, a one-step Newton update around the truth shows that the IVW estimator with optimal fixed weights coincides with the ML estimator up to the first order[2], and both are statistically efficient.

## 2.3  THEORETICAL ANALYSIS

We analyze the deviation of the complementary-label estimator $\widehat{A}_{\mathrm{comp}}$ under the uniform wrong-index assumption Eq. (1). Although $\widehat{A}_{\mathrm{comp}}$ is unbiased, its variance can be non-negligible at small sample sizes. We present two finite-sample deviation bounds: a distribution-free Hoeffding bound (Boucheron et al., 2003) and a data-dependent Bernstein bound (Maurer & Pontil, 2009) leveraging empirical variance. We then extend these to a fixed-weight mixture estimator.

### 2.3.1  BOUNDS FOR $\widehat{A}_{\mathrm{comp}}$

We derive a unified bound for $\widehat{A}_{\mathrm{comp}}$ by taking the minimum of the Hoeffding and empirical Bernstein inequalities. This formulation is both variance-free (via Hoeffding) and variance-adaptive (via Bernstein).The proof is shown in Appendix E.

**Theorem 2.** *With probability at least* $1 - \delta$,

$$\left| \widehat{A}_{\mathrm{comp}} - A \right| \leq (K-1)\,\min\left\{ \sqrt{\frac{\log(2/\delta)}{2\,n_{\mathrm{c}}}},\ \sqrt{\frac{2\,\hat{q}(1-\hat{q})}{n_{\mathrm{c}}-1}\,\log\frac{4}{\delta}} + \frac{7\,\log(4/\delta)}{3\,(n_{\mathrm{c}}-1)} \right\}.$$

The first branch (Hoeffding) is simple and variance-free but conservative. The second branch (empirical Bernstein) adapts to the empirical variance $\hat{q}(1 - \hat{q})$ and is typically tighter when $\hat{q}$ is close to 0 or 1. Both bounds require only i.i.d. sampling of $\{W_i\}$ under Eq. (1), and hold for any $n_{\mathrm{c}}$.

### 2.3.2  BOUNDS FOR $\hat{A}_{\mathrm{mix}}$

Consider the mixture estimator with a (possibly data-dependent) weight $w \in [0,1]$: $\widehat{A}_{\mathrm{mix}} = w\,\widehat{A}_{\mathrm{ord}} + (1-w)\,\widehat{A}_{\mathrm{comp}}$ where $\widehat{A}_{\mathrm{ord}} = \frac{1}{n_{\mathrm{o}}}\sum_{j=1}^{n_{\mathrm{o}}} z_j$, and $z_j \in \{0,1\}$. Since the ordinary and complementary sets are disjoint, we have $\widehat{A}_{\mathrm{mix}} - A = w(\widehat{A}_{\mathrm{ord}} - A) + (1-w)(\widehat{A}_{\mathrm{comp}} - A)$. With a union bound and Hoeffding inequality, we can derive the following proposition.

---

[2]A detailed proof is provided in Appendix D.

**Proposition 3.** *For any split* $\delta = \delta_{\mathrm{o}} + \delta_{\mathrm{c}}$, *with* $\delta_{\mathrm{o}}, \delta_{\mathrm{c}} > 0$, *with probability at least* $1 - \delta$,

$$
\left| \widehat{A}_{\mathrm{mix}} - A \right| \leq \min \left\{ \begin{array}{l} w \sqrt{\frac{\log(2/\delta_{\mathrm{o}})}{2\, n_{\mathrm{o}}}} + (1-w)(K-1) \sqrt{\frac{\log(2/\delta_{\mathrm{c}})}{2\, n_{\mathrm{c}}}}, \\[2mm] w \left( \sqrt{\frac{2\, \widehat{A}_{\mathrm{ord}}(1-\widehat{A}_{\mathrm{ord}})}{n_{\mathrm{o}}-1} \log \frac{4}{\delta_{\mathrm{o}}}} + \frac{7 \log(4/\delta_{\mathrm{o}})}{3\, (n_{\mathrm{o}}-1)} \right) \\[2mm] + (1-w) \left( \sqrt{\frac{2\, (K-1)^2\, \hat{q}(1-\hat{q})}{n_{\mathrm{c}}-1} \log \frac{4}{\delta_{\mathrm{c}}}} + \frac{7\, (K-1) \log(4/\delta_{\mathrm{c}})}{3\, (n_{\mathrm{c}}-1)} \right) \end{array} \right\}.
$$

*A symmetric choice* $\delta_{\mathrm{o}} = \delta_{\mathrm{c}} = \delta/2$ *yields the simpler logs* $\log(4/\delta)$ *and* $\log(8/\delta)$ *respectively.*

The union-bound approach provides guarantees that hold for *any* weight $w \in [0,1]$, including data-dependent plug-in choices such as $\widehat{w}_{\mathrm{IVW}}$. This makes it directly applicable in practice, although the constants are typically looser than those obtained from the moment generating function based Bernstein bound (See Theorem 4).

We now state a Bernstein-type PAC bound for the mixed estimator that *complements* the union-bound inequalities established in Proposition 3; its proof is deferred to Appendix F.

**Theorem 4.** *With probability at least* $1 - \delta$, *the following holds for* $\widehat{A}_{\mathrm{mix}}$:

$$
\left| \widehat{A}_{\mathrm{mix}} - A \right| \leq \sqrt{2v \log \frac{2}{\delta}} + c \log \frac{2}{\delta}, \tag{10}
$$

*where* $v = w^2 \frac{A(1-A)}{n_{\mathrm{o}}} + (1-w)^2 \frac{(K-1)^2\, q(1-q)}{n_{\mathrm{c}}}$, *and* $c = \max \left\{ \frac{w}{n_{\mathrm{o}}}, \frac{(1-w)(K-1)}{n_{\mathrm{c}}} \right\}$.

This bound tightens Hoeffding's inequality by incorporating both the variance term $v$ and the (range-driven) linear term $c$. It is valid for any fixed choice of the weight $w \in [0,1]$, for instance if $w$ is set as a constant or estimated on an independent pilot set. If instead $w$ is estimated from the same evaluation data (e.g. the plug-in $\widehat{w}_{\mathrm{IVW}}$), the fixed-weight assumption is violated; in that case one must either revert to the union-bound bounds (Proposition 3), or apply standard remedies such as sample splitting (Chernozhukov et al., 2018; Wager & Athey, 2018)or a finite grid with a union bound (Boucheron et al., 2013; Shalev-Shwartz & Ben-David, 2014). For practical plug-in forms and further discussion, see Appendix G.

## 3 EXPERIMENTS

We evaluate our unbiased and mixed estimators in three settings: **(I) Statistical validation** on standard multiple-choice benchmarks, where we verify unbiasedness and variance properties (see Eqs. (2), (4), (16)); **(II) Practical evaluation** on two real-world classification benchmarks: the Japanese financial dataset EDINET-Bench (Sugiura et al., 2026), where each option corresponds to a professional industry domain and annotators provide weak complementary feedback; and the English Medical Abstracts dataset (Schopf et al., 2023), which covers multiple disease categories with natural clinical text. **(III) Agentic training** in agent-based workflows, where we replace ordinary accuracy with our proposed estimator as the fitness function and examine the resulting agent performance. For agent scaffolding, we use ADAS (Hu et al., 2025) and AFlow (Zhang et al., 2025b). For details of the labeling protocol, please refer to Appendix C.

### 3.1 STATISTICAL VALIDATION OF COMPLEMENTARY AND MIXED ESTIMATORS

Statistical validation is conducted on four benchmarks. Since oracle labels are available, complementary labels are synthetically constructed following Appendix C. We report three types of evaluation metrics. **(A)** Unbiasedness and variance are examined by subsampling 300 ordinary and 300 complementary labels (120 for GPQA due to dataset size). Accuracy is estimated using Eq. (2), and variability is quantified through standard deviations, reported as $\widehat{A} \pm \widehat{\sigma}$ with $\widehat{\sigma}$ denoting the estimated standard deviation. **(B)** The relative efficiency of complementary labels is assessed by computing the number of complementary samples required to achieve the same variance as $n_{\mathrm{o}}$ ordinary labels, according to Eq. (4), where the full ordinary dataset is used to approximate the ground-truth accuracy $A$. **(C)** To mimic practical annotation protocols (Appendix C), we assume that each ordinary

label can be replaced by $(K - 1)$ complementary labels, where $K$ is the number of answer options. This setting is used to evaluate the mixed estimators in Eq. (7) and Eq. (9).

Each dataset is evaluated on all five metrics using `gpt-5-nano` (OpenAI, 2025b), with three independent runs averaged; aggregated outcomes are summarized in Table 1. For the MATH-MC benchmark, we use `GPT-4.1-nano` instead of `GPT-5-nano`. This choice is due to (i) output-format constraints in `GPT-5-nano` that conflict with our final-answer extraction protocol, and (ii) the need to avoid ceiling effects: in our pilot runs, `GPT-5-nano` attains near-ceiling accuracy on MATH-MC, which compresses the observable gaps among estimators.

Table 1: Performance of estimators across benchmarks. Values are reported as mean accuracy $\pm$ standard deviation across three random seeds; the average of per-run standard deviations is additionally shown in parentheses. A brief description of the estimators is provided in the legend below Detailed estimator setups and confidence-interval analyses are provided in Appendix I. The last column reports the macro-average across datasets (counting Math and Math-COT separately). For each benchmark, the estimator whose mean accuracy is closest to the Ord-Eval reference is highlighted in bold. Column-wise deviations from the oracle ($\Delta$) are reported in Appendix L.

| Estimator | MMLU-Pro | MedQA-USMLE | GPQA | MATH† | MATH(CoT)‡ | Average |
|---|---|---|---|---|---|---|
| Ord | 78.33 ± 1.73 (±2.38) | 92.89 ± 1.35 (±1.48) | 64.17 ± 1.67 (±4.39) | 47.56 ± 3.91 (±2.88) | 84.89 ± 0.77 (±2.07) | 73.57 |
| Comp-$n_o$ | 77.00 ± 12.49 (±7.95) | **92.67 ± 1.53 (±2.67)** | **59.17 ± 3.82 (±9.42)** | 48.44 ± 10.78 (±7.72) | 80.44 ± 2.78 (±4.98) | 71.54 |
| Comp-Var | 75.67 ± 2.15 (±2.51) | 90.61 ± 1.43 (±1.69) | 63.67 ± 5.01 (±4.28) | 41.10 ± 3.17 (±2.93) | 81.35 ± 0.29 (±2.28) | 70.48 |
| IVW-0.5 | 77.89 ± 1.58 (±1.80) | 91.61 ± 1.11 (±1.15) | 65.28 ± 1.34 (±3.33) | 43.44 ± 3.95 (±2.52) | 83.56 ± 1.17 (±1.58) | 72.36 |
| IVW | **77.97 ± 1.58 (±1.79)** | 91.86 ± 1.11 (±1.13) | 65.14 ± 1.38 (±3.30) | 44.87 ± 3.82 (±2.37) | **83.86 ± 0.83 (±1.56)** | 72.74 |
| ML | 77.94 ± 1.58 (±1.79) | 91.65 ± 1.08 (±1.18) | 65.11 ± 1.38 (±3.28) | **44.75 ± 3.79 (±2.36)** | 83.65 ± 1.04 (±1.59) | 72.62 |
| Ord-Eval | 77.97 | 92.66 | 59.52 | 44.21 | 83.89 | – |

**Legend:**
Ord = ordinary-label estimator; Comp-$n_o$ = complementary labels with the same sample size as ordinary; Comp-Var = complementary labels with variance-matched sample size; IVW-0.5 = fixed 0.5/0.5 weighting of ordinary and complementary estimates; IVW = inverse-variance–weighted combination with plug-in weights; ML = joint maximum-likelihood estimator using both label types; Ord-Eval = oracle accuracy on the full dataset.

Table 1 summarizes the results across benchmarks, together with a reference evaluation using all available ordinary labels (Ord-Eval). Importantly, higher accuracy in isolation does not imply a better estimator; the desideratum is to approximate the Ord-Eval reference as closely and reliably as possible under scarce supervision. With finite ordinary labels and $(K - 1)$-times complementary labels (where $K$ is the number of choices), Ord and Comp-$n_o$ can occasionally yield point estimates that are numerically closer to Ord-Eval (as seen in MedQA and GPQA). However, their standard errors are substantially wider, indicating poor reliability and limited reproducibility. For Comp-Var, increasing the number of complementary labels reduces variance, consistent with Eq. (3) and Eq. (4). We report both across-seed variability and the average within-run standard deviation, the latter more directly reflecting estimator variance. Ord-Eval itself should not be regarded as the ground-truth accuracy—especially on GPQA, where dataset size is small and uncertainty remains substantial—but rather as an empirical reference.

Mixture estimators (IVW and ML) reduce the gap to the Ord-Eval reference compared with both ordinary-only and complementary-only estimators across most benchmarks. Among mixture estimators, IVW consistently improves over the equal-weighted baseline in within-run standard deviation (IVW-0.5), confirming that inverse-variance weighting provides the optimal linear combination of ordinary and complementary labels. ML further achieves narrower intervals owing to its joint likelihood formulation. However, ML requires more precise pilot estimates of accuracy and may thus be less robust in practice, whereas IVW remains stable by directly incorporating variance esti-

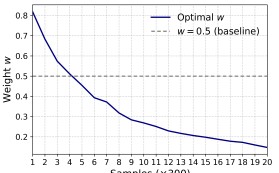
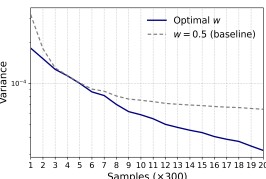

(a) Optimal $w$ vs. number of samples.

(b) Corresponding variance on a log scale.

Figure 2: Comparison of optimal weighting and baseline ($w = 0.5$). Optimal weighting consistently achieves lower variance compared to the equal-weighted baseline, demonstrating more reliable estimation with limited samples.

mates. This distinction is supported by the within-run standard deviations, which indicate that IVW achieves competitive efficiency with simpler computation.

Table 2: Mean acc. $\pm$ standard deviation across 3 seeds, w/ the average of per-run standard deviations. Bold: Lowest variance across 3 runs. See App. I for details.

| Estimator | EDINET | EDINET (Extended) | Medical Abstract |
|---|---|---|---|
| Ord | $19.35 \pm 5.59$ ($\pm 7.13$) | $21.67 \pm 2.46$ ($\pm 2.89$) | $70.38 \pm 0.62$ ($\pm 0.96$) |
| Comp-(K-1) | $10.75 \pm 8.12$ ($\pm 16.46$) | $26.60 \pm 3.45$ ($\pm 5.86$) | $68.79 \pm 0.71$ ($\pm 1.13$) |
| IVW-0.5 | $15.05 \pm 5.18$ ($\pm 8.98$) | $24.14 \pm 2.26$ ($\pm 3.27$) | $69.58 \pm 0.57$ ($\pm 0.74$) |
| IVW | $17.95 \pm 4.88$ ($\pm 6.53$) | $22.64 \pm 2.27$ ($\pm 2.59$) | $\mathbf{69.71 \pm 0.56}$ ($\mathbf{\pm 0.73}$) |
| ML | $\mathbf{17.93 \pm 4.87}$ ($\mathbf{\pm 6.29}$) | $\mathbf{22.61 \pm 2.28}$ ($\mathbf{\pm 2.26}$) | $69.70 \pm 0.56$ ($\pm 0.74$) |

Overall, while Comp-$n_{\mathrm{o}}$ may occasionally appear closest to Ord-Eval in mean accuracy, its instability makes it unreliable for reproducible evaluation. IVW and ML instead strike the best balance between bias and variance, yielding stable and reproducible estimates close to the Ord-Eval reference across datasets. The empirical closeness of the two methods further validates our theoretical analysis, suggesting that mixture estimators can serve as a practical foundation for scalable oversight in settings where ground-truth labels are scarce or infeasible.

**Ablation Study on IVW.** Recall that the optimal coefficient for IVW is given in Eq. (8). When setting $n_{\mathrm{c}} = (K - 1)n_{\mathrm{o}}$ (where $K$ denotes the number of choices), the expression simplifies to $\frac{A+K-2}{AK+K-2} = \frac{1}{K}\left(1 + \frac{(K-1)(K-2)}{AK+K-2}\right)$. Since $0 \leq A \leq 1$, the coefficient lies in the range $\frac{1}{2} \leq \frac{1}{K}\left(1 + \frac{(K-1)(K-2)}{AK+K-2}\right) \leq 1$. In particular, when $A$ is close to 1—as suggested by the Ord-Eval reference—the optimal weight approaches $\frac{1}{2}$, explaining why in Table 1 the IVW estimator often performs similarly to the ordinary baseline.

To further examine whether the closed-form coefficient yields the optimal trade-off, we conduct an ablation on **MedQA-USMLE** by varying the number of complementary labels from $n_{\mathrm{c}} = n_{\mathrm{o}}$ to $n_{\mathrm{c}} = 20n_{\mathrm{o}}$ (with $n_{\mathrm{o}} = 300$). Results are averaged over three random seeds. As shown in Figure 2, when $n_{\mathrm{c}} \approx 3n_{\mathrm{o}}$ (the setting used in Table 1), the optimal weight is close to the fixed baseline $w = 0.5$. However, for $n_{\mathrm{c}} < n_{\mathrm{o}}$ or $n_{\mathrm{c}} > 6n_{\mathrm{o}}$, the optimal variance is consistently lower than the baseline. This behavior is intuitive: once the number of complementary labels exceeds the variance-matching threshold in Eq. (4), the complementary estimator alone has smaller variance, and the closed-form IVW solution naturally shifts weight away from ordinary labels. By contrast, the fixed baseline cannot adapt, leading to consistently higher variance.

## 3.2 BENCHMARK EVALUATION IN REAL-WORLD TASKS

To demonstrate the practicality of our method in real-world scenarios, we conduct proof-of-concept experiments for *estimation and learning under partitioned human supervision*, as directly collecting superhuman tasks would be prohibitively costly. Following the data collection protocol in Algorithm 1, for each domain we randomly select an expert and ask whether a given object belongs to their field. Applied to the benchmarks, complementary labels are generated from the oracle labels by splitting each instance into 1 ordinary label and $K - 1$ complementary labels, corresponding to the $K$ candidate classes.

We evaluate on two benchmarks. First, we use EDINET-Bench (Sugiura et al., 2026), a Japanese financial report classification dataset, where each industry corresponds to a professional domain, and we adopt the prompting setup of Sugiura et al. (2026). Since the original dataset contains only 496 samples (comparable to GPQA), we extend it following their released code, resulting in 3,269 samples; we refer to this as EDINET-Bench Extended. To our knowledge, there are no public medical benchmarks in which each class explicitly corresponds to a medical specialization. We therefore adopt the Medical Abstracts dataset (Schopf et al., 2023), which consists of paper abstracts labeled by disease categories, simulating the task of narrow domain experts classifying diseases.

Both EDINET-Bench Extended and Medical Abstracts are imbalanced across classes. Nevertheless, our assumption in Eq. (1) is that complementary labels are sampled uniformly at random, independent of the original label distribution. We follow this procedure and empirically verify that class imbalance does not affect the validity of our estimator.

**Results** As shown in Table 2, both IVW and ML consistently achieve the lowest average per-run standard deviation, confirming the variance reduction effect observed in § 3.1. In contrast, the Comp-(K-1) estimator exhibits much larger variance on EDINET-Bench, which aligns with Eq. (4): with $K = 16$ industries and relatively low accuracy, the variance is amplified.

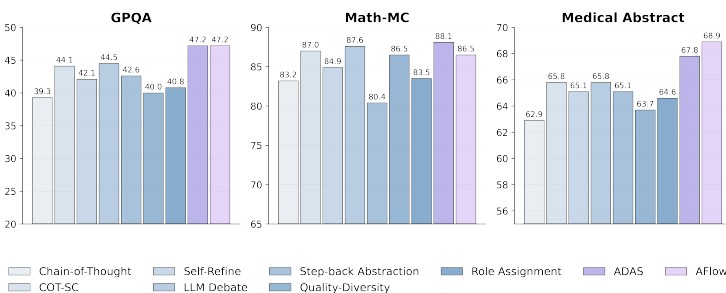

Figure 3: Accuracy across different benchmarks. Across GPQA, Math-MC, and Medical Abstract, agentic systems guided by weak complementary signals (ADAS and AFlow) consistently outperform manually designed baselines.

Comparing EDINET-Bench and its extended version, we observe that increasing the number of samples substantially reduces the fluctuations of the estimators, supporting our earlier finding in § 3.1 that small-scale benchmarks such as GPQA can lead to unstable estimates. Finally, on the Medical Abstracts dataset, IVW and ML again achieve the most stable performance, demonstrating that the complementary-label estimators generalize across domains and languages.

### 3.3 AGENT SEARCH WITH ESTIMATOR FEEDBACK

Finally, we show that weak human supervision can be leveraged not only for evaluation but also as training signal for AI systems. We evaluate agentic methods under complementary-label signals on three benchmarks. GPQA serves as the most challenging setting in our study (Table 1); Math-MC focuses on reasoning ability; and Medical Abstracts best captures our practical scenario. Following Hu et al. (2025), we include CoT (Wei et al., 2022), CoT-SC (Wang et al., 2023), Self-Refine (Madaan et al., 2023), LLM Debate (Du et al., 2024), Step-back Abstraction (Zheng et al., 2024), Quality-Diversity (Lu et al., 2025), and Role Assignment (Xu et al., 2025) as manually designed baselines, and compare against ADAS (Hu et al., 2025) and AFlow (Zhang et al., 2025b). We define the accuracy computed by $q$ as *complementary accuracy* and apply Eq. (2) to obtain the *transformed complementary accuracy*. For ADAS and AFlow, we follow the original protocol (see Appendix J), computing both complementary and transformed complementary accuracy on the validation set and reporting the better-performing variant.

Figure 3 compares ADAS and AFlow against strong baselines on GPQA, Math-MC, and Medical Abstracts. Both agentic systems consistently outperform manually designed workflows once weak complementary-label signals are incorporated, demonstrating that weak human feedback can provide learning signal. While the choice of estimator (raw vs. transformed complementary accuracy) leads to different outcomes depending on task difficulty, the overall pattern remains robust: weak signals suffice to improve performance beyond established baselines, with additional variance analysis provided in Appendix K. To further illustrate the resulting agentic system's behavior, we present a case study in Appendix M. These findings reinforce the central theme of our study: humans' weak labels can provide a reliable and scalable supervisory channel for steering agentic AI systems.

## 4 RELATED WORK

**Scalable oversight** In superhuman regimes where AI systems surpass human abilities, humans can no longer provide reliable supervision to evaluate and train them. Recent scalable-oversight work explores strategies that let limited (human) supervisors steer stronger models. Among the many scalable oversight protocols (such as weak-to-strong generalization (Burns et al., 2024), easy-to-hard generalization (Sun et al., 2024), debate (Khan et al., 2024), and RLAIF (Bai et al., 2022)), our approach may seem similar to iterated amplification/IDA (Christiano et al., 2018) and recursive task decomposition (Wu et al., 2021) at a glance. The difference is that our approach can be used for tasks that are not easily decomposable, as we are not decomposing the task, but rather relying on expertise-wise partitioned human supervision to enable the usage of complementary labels.

**Weak supervision** Weakly-supervised learning (Sugiyama et al., 2022) studies learning with weaker form of supervision instead of ground-truth labels. Complementary labels (Ishida et al., 2017; 2019; Yu et al., 2018; Wang et al., 2024a; 2025), in particular, provide "not-this-class" in-

formation, with the traditional motivation of reducing annotation cost. Our perspective differs: we leverage weak supervision not merely for cheaper labeling, but as a principled route to scalable oversight in the superhuman regime, where no single human expert can provide verification end-to-end.

**Agentic systems**   LLMs have inspired a wide range of system architectures (Wei et al., 2022; Wang et al., 2023; Madaan et al., 2023; Shinn et al., 2023), in particular by incorporating multiple LLMs into collaborative frameworks (Du et al., 2024; Liu et al., 2024; Chen et al., 2024; Wu et al., 2024; Wang et al., 2024c). Early studies relied on manually designed agentic workflows, which demanded substantial manual design and domain expertise. More recent work has shifted toward *automated agent discovery*, in which LLMs themselves are used to generate, refine, and optimize workflow structures (Hu et al., 2025; Zhang et al., 2025a; 2026; 2025b), but required ground-truth supervision to construct a fitness score. We investigate whether automated agentic system design can proceed even with weaker training signals, i.e., complementary labels.

## 5   Discussion & Limitations

**Assumptions and extensions**   The assumption of uniformly querying an index plays a key role in obtaining unbiased complementary labels, and in our labeling protocol this assumption is guaranteed for the practical scenario we consider. However, to make the whole system more robust, we should also consider possible issues that may arise in practice, such as biased answers from experts, uncertain answers (which can be reframed as a type of noise), and overlaps between expert fields. For these cases, a wide range of complementary-label variants has been studied, such as biased complementary labels (Yu et al., 2018; Wang et al., 2024a), noisy complementary labels (Ishiguro et al., 2022), and multiple complementary labels (Feng et al., 2020). We believe that incorporating these formulations would further enhance the robustness of our framework.

Beyond the assumptions on the labels themselves, we also need to consider the structural assumption that the task is posed in a multiple-choice format. The multiple-choice format still has wide application, and in scenarios beyond the standard multiple choice setting, some open-ended questions can be transformed into structured multiple-choice forms to allow the use of our framework. For example, in debugging, modern systems strongly rely on cooperation among different sectors. The end-to-end service may look open-ended, but it can often be separated into different sectors or a hierarchical structure. In particular, when the root cause is unknown, we can ask engineers responsible for different parts of the product to provide weak signals, for example, confirming that the bug does not originate from their component. This kind of knowledge fragmentation provides opportunities to apply our framework.

**Limitations**   Although our framework has shown its potential for addressing the expertise problem in evaluating foundation models, we would still like to discuss its current limitations. First, the framework assumes the availability of complementary labels. When even complementary labels are scarce or inapplicable, for example, in open-ended problems that are hard to decompose into a multiple-choice pipeline, complementary labels may be ill-defined, and the formulation may need to be revised. Second, when feedback from humans is no longer available and the boundary of the ground truth becomes ambiguous, neither complementary labels nor ordinary labels can be reliably defined. In such cases, new measures of accuracy would be required to revise the framework. We view these current limitations as interesting directions for future research.

## 6   Conclusion

We introduced *scalable oversight via partitioned human supervision*, a protocol that utilizes the weak human supervision for tasks beyond any single expert's ability. We derived an unbiased estimator of performance metric from complementary labels, analyzed its variance, and provided finite-sample guarantees. We further proposed two practical mixture estimators that combine scarce ordinary labels with abundant complementary labels. Empirically, we show that these estimators allow us to estimate the performance of AI systems without requiring full ground truth. They also serve as effective training signals which we demonstrate with automatically designing agentic workflows.

## ACKNOWLEDGEMENTS

RY was supported by the RIKEN-AIP Undergraduate Research Program, TI was supported by JST ASPIRE Grant Number JPMJAP25B1, and MS was supported by JST ASPIRE Grant Number JP-MJAP2405.

## ETHICS STATEMENT

Our work is motivated by AI safety: we aim to improve oversight of advanced AI systems by enabling evaluation and training even on tasks beyond any single human expert's competence. This approach is intended as a step toward improving the reliability and alignment of future AI models with human interests, which we believe is a positive contribution to society.

Currently, we do not foresee immediate direct negative social impacts from the research itself. However, we recognize that alignment and oversight techniques can be a double-edged sword: they are dual-use in the sense that they could be repurposed by malicious actors to optimize systems toward harmful objectives (Machine Learning Street Talk, 2025), and even when used defensively, they can be vulnerable to strategic behavior that produces a false sense of safety (Hubinger et al., 2024; Greenblatt et al., 2024). These risks are not unique to our approach, but we caution against such misuse and vulnerabilities.

## REPRODUCIBILITY STATEMENT

Code and scripts to reproduce our experiments are available in our public GitHub repository: 
Towards Scalable Oversight via Partitioned Human Supervision.

For ADAS (Hu et al., 2025) and AFlow (Zhang et al., 2025b), we follow their official implementations; see Appendix J for additional details. Because our experiments use the OpenAI API, exact replication may vary over time if models are updated or retired. We therefore provide full implementation details, dataset references, and evaluation scripts to support reproducibility as closely as possible.

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

## A    PROOF OF COROLLARY 1

*Proof.* For a single item, under Eq. (1),

$$\mathbb{E}[W] = \Pr(\hat{Y} = Y) \Pr(W = 1 \mid \hat{Y} = Y) + \Pr(\hat{Y} \neq Y) \Pr(W = 1 \mid \hat{Y} \neq Y)$$

$$= \Pr(\hat{Y} = Y) \cdot 1 + \Pr(\hat{Y} \neq Y) \cdot \left(1 - \tfrac{1}{K-1}\right) = \frac{A + K - 2}{K - 1}.$$

If we consider $\hat{q} = \frac{1}{n_c} \sum_{i=1}^{n_c} w_i$, an empirical version of $\mathbb{E}[W]$, this $\hat{q}$ is an unbiased estimator of $\mathbb{E}[W]$. Hence, $\widehat{A}_{\mathrm{comp}} = (K-1)\hat{q} - (K-2)$ is an unbiased estimator of $A$. □

## B    CONNECTION BETWEEN THE COMPLEMENTARY-LABEL ESTIMATOR AND COMPLEMENTARY LOSS

**Notation.**    Let $(X, Y) \sim \mathcal{D}$ with $Y \in [K] = \{1, \ldots, K\}$. Given a model $g$, denote its prediction by $\hat{Y} = g(X)$. For any class $k \in [K]$, the loss is $\ell(k, \hat{Y})$. In complementary-label learning (Ishida et al., 2019), one observes $\bar{Y} \neq Y$ sampled uniformly from $[K] \setminus \{Y\}$. The complementary loss is defined (Eq. (10) in (Ishida et al., 2019)) as

$$\bar{\ell}(k, \hat{Y}) = -(K-1)\,\ell(k, \hat{Y}) + \sum_{j=1}^{K} \ell(j, \hat{Y}).$$

**Specialization to 0-1 loss.**    For the 0–1 loss $\ell(k, \hat{Y}) = \mathbb{I}\{\hat{Y} \neq k\}$, we compute

$$\begin{aligned}
\bar{\ell}(k, \hat{Y}) &= -(K-1)\,\mathbb{I}\{\hat{Y} \neq k\} + \sum_{j=1}^{K} \mathbb{I}\{\hat{Y} \neq j\} \\
&= -(K-1)\,\mathbb{I}\{\hat{Y} \neq k\} + (K-1) \\
&= (K-1)\,\mathbb{I}\{\hat{Y} = k\}.
\end{aligned} \tag{11}$$

**Risk under complementary labels.**    The standard risk is

$$R = \mathbb{E}_{(X,Y)}[\ell(Y, \hat{Y})] = \Pr(\hat{Y} \neq Y) = 1 - A,$$

where $A = \Pr(\hat{Y} = Y)$ is the accuracy. By the risk-rewrite identity of Ishida et al. (2019) (Eq. (8)), we also have

$$R = \mathbb{E}_{(X,\bar{Y})}[\bar{\ell}(\bar{Y}, \hat{Y})] = (K-1)\,\Pr(\hat{Y} = \bar{Y}).$$

**Equivalence to our estimator.**    Let $W = \mathbb{I}\{\hat{Y} \neq \bar{Y}\}$. Then $\Pr(\hat{Y} = \bar{Y}) = 1 - \mathbb{E}[W]$, so

$$1 - A = (K-1)\,(1 - \mathbb{E}[W]).$$

Rearranging yields

$$A = (K-1)\,\mathbb{E}[W] - (K-2).$$

This is exactly the linear correction in Eq. (2). Replacing $\mathbb{E}[W]$ by its empirical mean $\hat{q} = \frac{1}{n_c} \sum_{i=1}^{n_c} w_i$ gives the unbiased estimator $\widehat{A}_{\mathrm{comp}} = (K-1)\hat{q} - (K-2)$ of Theorem 1.

## C    DATA LABELING AND ESTIMATION PROTOCOL

In multiple-choice evaluation, the effective "class" is the *index position* of an option rather than its textual content. To satisfy the uniformity condition in Eq. (1), the labeling process proceeds as follows. For each item, all options are first shuffled uniformly at random. After shuffling, the correct index $y$ is recorded. A complementary label is then created by sampling a wrong index $\bar{y}$ uniformly from the $K - 1$ remaining positions. The final published datum is $(x, \text{shuffled options}, \bar{y})$. When

the ground truth is known, this procedure can be automated: $y$ is only used internally for generating $\bar{y}$, and the released dataset never exposes $y$. When labels are collected from human raters, the annotation interface can enforce the same randomization in the backend to guarantee uniformity. Any deviation, such as asking raters to pick the "most plausible wrong option," would violate the assumption behind Eq 1.

The same protocol can be extended to cases where the ground truth is not available in advance. A complementary label is first generated in the same way, and then the annotator is asked to verify whether it happens to be correct. This verification yields a mixture of ordinary and complementary labels, which forms the mixed dataset studied in § 2.2.

The end-to-end evaluation pipeline, including both ordinary and complementary sets and the associated estimators, is summarized in Algorithm 1. Two estimation strategies are supported: a closed-form maximum-likelihood estimator and an inverse-variance weighted (IVW) estimator.

---

**Algorithm 1** Collecting complementary labels and estimating accuracy (fixed $K$)

---

1: **Input:** MCQ items; evaluation model $f$; sizes $n_{\mathrm{o}}, n_{\mathrm{c}}$.
2: **for** each item **do**
3:     Shuffle options uniformly; record correct index $y$.
4:     **if** item $\in$ ordinary set **then**
5:         Evaluate $f$ to obtain prediction $\hat{y}$; record $z = \mathbb{I}\{\hat{y} = y\}$.
6:     **else**                                                                       ▷ complementary set
7:         Sample $\bar{y}$ uniformly from $\{1, \ldots, K\} \setminus \{y\}$; publish $(x, \text{options}, \bar{y})$.
8:         Evaluate $f$ to obtain prediction $\hat{y}$; record $W = \mathbb{I}\{\hat{y} \neq \bar{y}\}$.
9:     **end if**
10: **end for**
11: Compute $S_{\mathrm{o}} = \sum_j z_j$, $S_{\mathrm{c}} = \sum_i w_i$.
12: **Option A (ML):** return $\widehat{A}_{\mathrm{ML}}$ via Eq. (9) and a confidence interval from observed information.
13: **Option B (IVW):** compute $\widehat{A}_{\mathrm{ord}}$, $\widehat{A}_{\mathrm{comp}}$, and $\widehat{w}$, then return $\widehat{A}_{\mathrm{IVW}}$ with plug-in confidence interval.

---

Two assumptions are critical for the validity of these estimators. First, the wrong index in each complementary label must be sampled uniformly; systematic biases, such as consistently preferring hard negatives, lead to biased estimates of $\widehat{A}_{\mathrm{comp}}$. Second, the ordinary and complementary sets must be drawn from the same distribution of items. If domain shift or subject imbalance exists, the combined estimator reflects accuracy on a mixture distribution. In such cases, stratified or subject-wise weighting may be preferable.

## D   ONE-STEP NEWTON UPDATE AND IVW EQUIVALENCE

**Setup and notation.**   Let $A \in (0, 1)$ denote the (unknown) accuracy for a $K$-way multiple-choice task ($K \geq 3$). We observe two independent samples: (i) ordinary labels $S_{\mathrm{o}} \sim \mathrm{Bin}(n_{\mathrm{o}}, A)$ with empirical accuracy $\widehat{A}_{\mathrm{ord}} = S_{\mathrm{o}}/n_{\mathrm{o}}$; (ii) complementary labels obtained by uniformly sampling a wrong option per item, so that $S_{\mathrm{c}} \sim \mathrm{Bin}(n_{\mathrm{c}}, q)$ with

$$q(A) = \frac{A + K - 2}{K - 1}, \qquad \widehat{q} = S_{\mathrm{c}}/n_{\mathrm{c}}, \qquad \widehat{A}_{\mathrm{comp}} = (K-1)\widehat{q} - (K-2).$$

We consider a pilot $A_0$ (e.g. $\widehat{A}_{\mathrm{ord}}$ or $\widehat{A}_{\mathrm{comp}}$) and write $q_0 = q(A_0)$.

**Newton update and plug-in simplification.**   The joint log-likelihood is

$$\ell(A) = S_{\mathrm{o}} \log A + (n_{\mathrm{o}} - S_{\mathrm{o}}) \log(1 - A) + S_{\mathrm{c}} \log q + (n_{\mathrm{c}} - S_{\mathrm{c}}) \log(1 - q).$$

Differentiating gives the score functions

$$U_{\mathrm{o}}(A) = \frac{n_{\mathrm{o}}(\widehat{A}_{\mathrm{ord}} - A)}{A(1 - A)}, \qquad U_{\mathrm{c}}(A) = \frac{n_{\mathrm{c}}(\widehat{A}_{\mathrm{comp}} - A)}{(K - 1)^2 \, q(1 - q)}.$$

The observed information terms are

$$\mathcal{I}_{\mathrm{o}}^{\mathrm{obs}}(A) = \frac{S_{\mathrm{o}}}{A^2} + \frac{n_{\mathrm{o}} - S_{\mathrm{o}}}{(1 - A)^2}, \qquad \mathcal{I}_{\mathrm{c}}^{\mathrm{obs}}(A) = \frac{1}{(K-1)^2} \left( \frac{S_{\mathrm{c}}}{q^2} + \frac{n_{\mathrm{c}} - S_{\mathrm{c}}}{(1 - q)^2} \right).$$

A one-step Newton update around $A_0$ is

$$A_1 = A_0 - \frac{U(A_0)}{U'(A_0)} = A_0 + \frac{U_{\mathrm{o}}(A_0) + U_{\mathrm{c}}(A_0)}{\mathcal{I}_{\mathrm{o}}^{\mathrm{obs}}(A_0) + \mathcal{I}_{\mathrm{c}}^{\mathrm{obs}}(A_0)}.$$

When we set $A_0 = \widehat{A}_{\mathrm{ord}}$, the ordinary part simplifies exactly: since $S_{\mathrm{o}} = n_{\mathrm{o}} A_0$, we have $U_{\mathrm{o}}(A_0) = 0$ and $\mathcal{I}_{\mathrm{o}}^{\mathrm{obs}}(A_0) = n_{\mathrm{o}} / [A_0(1 - A_0)] = I_{\mathrm{o}}(A_0)$, the Fisher information. For the complementary part, we plug in $q = \widehat{q}$ so that $S_{\mathrm{c}} = n_{\mathrm{c}} \widehat{q}$ holds identically. This yields

$$\mathcal{I}_{\mathrm{c}}^{\mathrm{obs}}(A_0)\big|_{q=\widehat{q}} = \frac{n_{\mathrm{c}}}{(K-1)^2 \, \widehat{q}(1 - \widehat{q})} = I_{\mathrm{c}}(\widehat{q}), \qquad U_{\mathrm{c}}(A_0)\big|_{q=\widehat{q}} = I_{\mathrm{c}}(\widehat{q}) \, (\widehat{A}_{\mathrm{comp}} - A_0).$$

Substituting back into the Newton update gives

$$A_1 \approx A_0 + \frac{I_{\mathrm{c}}(\widehat{q})(\widehat{A}_{\mathrm{comp}} - A_0)}{I_{\mathrm{o}}(A_0) + I_{\mathrm{c}}(\widehat{q})}.$$

**Closed-form IVW rule.** Rearranging shows that the practical one-step update is a convex combination of the two empirical estimators:

$$\widehat{A}_{\mathrm{IVW}} \approx \frac{I_{\mathrm{o}}(\widehat{A}_{\mathrm{ord}}) \, \widehat{A}_{\mathrm{ord}} + I_{\mathrm{c}}(\widehat{q}) \, \widehat{A}_{\mathrm{comp}}}{I_{\mathrm{o}}(\widehat{A}_{\mathrm{ord}}) + I_{\mathrm{c}}(\widehat{q})}.$$

**Variance–information identities and final form.** For the two constituent estimators, the variance–information identities are

$$\mathrm{Var}(\widehat{A}_{\mathrm{ord}}) = \frac{1}{I_{\mathrm{o}}}, \qquad \mathrm{Var}(\widehat{A}_{\mathrm{comp}}) = \frac{1}{I_{\mathrm{c}}},$$

with the plug-in Fisher information

$$I_{\mathrm{o}}(\widehat{A}_{\mathrm{ord}}) = \frac{n_{\mathrm{o}}}{\widehat{A}_{\mathrm{ord}}(1 - \widehat{A}_{\mathrm{ord}})}, \qquad I_{\mathrm{c}}(\widehat{q}) = \frac{n_{\mathrm{c}}}{(K-1)^2 \, \widehat{q}(1 - \widehat{q})}.$$

Therefore the IVW weight can be written either in information form, $w^\star = \frac{I_{\mathrm{o}}}{I_{\mathrm{o}} + I_{\mathrm{c}}}$, or, equivalently, in variance form (using $I_\bullet = 1/\mathrm{Var}(\widehat{A}_\bullet)$),

$$w^\star = \frac{I_{\mathrm{o}}}{I_{\mathrm{o}} + I_{\mathrm{c}}} = \frac{\mathrm{Var}(\widehat{A}_{\mathrm{comp}})}{\mathrm{Var}(\widehat{A}_{\mathrm{ord}}) + \mathrm{Var}(\widehat{A}_{\mathrm{comp}})}.$$

Plugging these into the convex combination yields exactly results in §2.2:

$$\widehat{A}_{\mathrm{IVW}} = w^\star \, \widehat{A}_{\mathrm{ord}} + (1 - w^\star) \, \widehat{A}_{\mathrm{comp}}.$$

In summary, the IVW estimator is a one-step Newton approximation to the joint ML: it is first-order equivalent and asymptotically efficient, but may differ in finite samples because the weights are fixed by plug-in information rather than optimized jointly. While the ML is theoretically optimal, the IVW rule is a simple linear combination that often delivers practically comparable performance and is easier to apply in practice.

## E   PROOF OF THEOREM 2

*Proof.* **Hoeffding part.** For i.i.d. $w_i \in [0, 1]$, Hoeffding gives

$$\Pr\big(|\hat{q} - \mathbb{E}\hat{q}| \geq \epsilon\big) \leq 2 \exp\big(-2 n_{\mathrm{c}} \epsilon^2\big).$$

Since $\widehat{A}_{\mathrm{comp}} = (K-1)\hat{q} - (K-2)$ and $A = (K-1)\mathbb{E}\hat{q} - (K-2)$, we have

$$|\widehat{A}_{\mathrm{comp}} - A| = (K-1)\,|\hat{q} - \mathbb{E}[\hat{q}]|.$$

**Empirical Bernstein part.** Apply the two-sided empirical Bernstein inequality (Maurer & Pontil, 2009) to $\hat{q} = \frac{1}{n_{\mathrm{c}}}\sum_{i=1}^{n_{\mathrm{c}}} w_i$ with $w_i \in [0,1]$:

$$\left|\hat{q} - \mathbb{E}\hat{q}\right| \;\leq\; \sqrt{\frac{2\,\hat{q}(1-\hat{q})\,\log(4/\delta)}{n_{\mathrm{c}}}} \;+\; \frac{7\,\log(4/\delta)}{3\,(n_{\mathrm{c}}-1)}.$$

Multiplying both sides by $(K-1)$ and using $\widehat{A}_{\mathrm{comp}} - (A) = (K-1)\big(\hat{q} - \mathbb{E}[\hat{q}]\big)$ yields the second branch in Theorem 2 $\hfill\square$

## F    PROOF OF THEOREM 4

*Proof.* Recall $z_j = \mathbb{I}\{\hat{y}_j = y_j\}$, $j = 1,\ldots,n_{\mathrm{o}}$, are i.i.d. Bernoulli($A$) with $\mathbb{E}[z_j] = A$; and $w_i = \mathbb{I}\{\hat{y}_i \neq \bar{y}_i\}$, $i = 1,\ldots,n_{\mathrm{c}}$, are i.i.d. Bernoulli($q$) with $q = (A + K - 2)/(K-1)$ under Eq. (1). The two groups are independent.

Define the rescaled variables

$$X_j^{(o)} := \tfrac{w}{n_{\mathrm{o}}}z_j, \qquad X_i^{(c)} := \tfrac{(1-w)(K-1)}{n_{\mathrm{c}}}w_i,$$

and collect them as $X_1,\ldots,X_n$ with $n = n_{\mathrm{o}} + n_{\mathrm{c}}$. Then the centered sum

$$S := \sum_{j=1}^{n_{\mathrm{o}}}(X_j^{(o)} - \mathbb{E}X_j^{(o)}) \;+\; \sum_{i=1}^{n_{\mathrm{c}}}(X_i^{(c)} - \mathbb{E}X_i^{(c)}) \;=\; \widehat{A}_{\mathrm{mix}} - A. \qquad (12)$$

We verify the conditions of Theorem 2.10 in Boucheron et al. (2003):

*(i) Independence)* holds by construction.

*(ii) Boundedness).* For all $j, i$,

$$\left|X_j^{(o)} - \mathbb{E}X_j^{(o)}\right| \leq \tfrac{w}{n_{\mathrm{o}}}, \qquad \left|X_i^{(c)} - \mathbb{E}X_i^{(c)}\right| \leq \tfrac{(1-w)(K-1)}{n_{\mathrm{c}}}.$$

Set

$$c := \max\left\{\tfrac{w}{n_{\mathrm{o}}},\ \tfrac{(1-w)(K-1)}{n_{\mathrm{c}}}\right\}. \qquad (13)$$

*(iii) Second moment).* Summing variances,

$$v := \sum_{j=1}^{n_{\mathrm{o}}}\mathrm{Var}(X_j^{(o)}) + \sum_{i=1}^{n_{\mathrm{c}}}\mathrm{Var}(X_i^{(c)}) = w^2\tfrac{A(1-A)}{n_{\mathrm{o}}} + (1-w)^2\tfrac{(K-1)^2\,q(1-q)}{n_{\mathrm{c}}}. \qquad (14)$$

*(iv) Higher moments).* Since $|X_i - \mathbb{E}X_i| \leq c$, for all integers $q \geq 3$, $(X_i - \mathbb{E}X_i)_+^q \leq c^{q-2}(X_i - \mathbb{E}X_i)^2$; summing and taking expectations yields $\sum_i \mathbb{E}[(X_i)_+^q] \leq v\,c^{q-2}$, which is dominated by the requirement $\frac{q!}{2}\,v\,c^{q-2}$ in (Boucheron et al., 2003, Thm. 2.10).

Therefore, Theorem 2.10 applies to $S$ in Eq. (12) and (using the symmetric lower tail plus a union bound) gives $\Pr\{|S| \geq \sqrt{2vt} + ct\} \leq 2e^{-t}$. Setting $t = \log(2/\delta)$ yields Eq. (10). $\hfill\square$

## G    PRACTICAL PLUG-IN FORMS OF THE BERNSTEIN BOUND

The quantities $A$ and $q$ are unknown; replace them by their empirical counterparts. Define
$\hat{v}_{\mathrm{o}} = \frac{\hat{A}_{\mathrm{ord}}(1-\hat{A}_{\mathrm{ord}})}{n_{\mathrm{o}}}$, $\hat{v}_{\mathrm{c}} = \frac{(K-1)^2\,\hat{q}(1-\hat{q})}{n_{\mathrm{c}}}$, $\hat{v}_{\mathrm{mix}} = w^2\hat{v}_{\mathrm{o}} + (1-w)^2\hat{v}_{\mathrm{c}}$, with $\hat{q} = S_{\mathrm{c}}/n_{\mathrm{c}}$.
Substituting into Eq. (10) yields the computable bound $\left|\hat{A}_{\mathrm{mix}} - A\right| \leq \sqrt{2\,\hat{v}_{\mathrm{mix}}\log\frac{2}{\delta}} + c\log\frac{2}{\delta}$, where $c$ is as in Eq. (13). For clarity, an equivalent compact statement is

$$\left|\hat{A}_{\mathrm{mix}} - A\right| \;\leq\; \sqrt{2\log\tfrac{2}{\delta}\,\big[w^2\hat{v}_{\mathrm{o}} + (1-w)^2\hat{v}_{\mathrm{c}}\big]} \;+\; \log\tfrac{2}{\delta}\,\max\!\left(\tfrac{w}{n_{\mathrm{o}}},\ \tfrac{(1-w)(K-1)}{n_{\mathrm{c}}}\right). \qquad (15)$$

The variance component is minimized at $w^\star = \frac{v_{\mathrm{c}}}{v_{\mathrm{o}}+v_{\mathrm{c}}}$, a practical $w$ is defined as $\hat{w}_{\mathrm{IVW}} = \frac{\hat{v}_{\mathrm{c}}}{\hat{v}_{\mathrm{o}}+\hat{v}_{\mathrm{c}}}$, coinciding with the IVW weight in Eq. (7). At this choice, the dominant (variance) term equals $\sqrt{2\log\frac{2}{\delta} \cdot \frac{v_{\mathrm{o}}v_{\mathrm{c}}}{v_{\mathrm{o}}+v_{\mathrm{c}}}}$ .

Defining $\widehat{h}_{\min} := \frac{\hat{v}_{\mathrm{o}}\hat{v}_{\mathrm{c}}}{\hat{v}_{\mathrm{o}}+\hat{v}_{\mathrm{c}}}$, using the fully plug-in version gives

$$|\hat{A}_{\mathrm{mix}} - A| \leq \sqrt{2\log\frac{2}{\delta}\,\widehat{h}_{\min}} + \log\frac{2}{\delta}\,\max\!\left(\frac{\hat{w}_{\mathrm{IVW}}}{n_{\mathrm{o}}}, \frac{(1-\hat{w}_{\mathrm{IVW}})(K-1)}{n_{\mathrm{c}}}\right), \qquad (16)$$

Conceptually, the mgf-based Bernstein bound (Eq. (10)) tightens Hoeffding's inequality by adding both a variance term $v$ and a range-driven linear term $c$. As $n_{\mathrm{o}}$ or $n_{\mathrm{c}}$ grows, the right-hand side vanishes: the variance part decays as $O(1/n_{\mathrm{o}}) + O(1/n_{\mathrm{c}})$, while the linear term scales as $O\!\big(\max\{w/n_{\mathrm{o}},\,(1-w)(K-1)/n_{\mathrm{c}}\}\big)$. The plug-in IVW weight $\widehat{w}_{\mathrm{IVW}} = \frac{\hat{v}_{\mathrm{c}}}{\hat{v}_{\mathrm{o}}+\hat{v}_{\mathrm{c}}}$ converges to $w^\star = \frac{v_{\mathrm{c}}}{v_{\mathrm{o}}+v_{\mathrm{c}}}$; in particular, $w^\star \to 0$ when $n_{\mathrm{c}} \to \infty$ with $n_{\mathrm{o}}$ fixed (the complementary arm dominates), and $w^\star \to 1$ when $n_{\mathrm{o}} \to \infty$ with $n_{\mathrm{c}}$ fixed (the ordinary arm dominates).

Formally, Eq. (10) is valid for any *fixed* choice of the weight $w \in [0,1]$ (e.g., a constant choice or an estimate obtained from an independent pilot split). If instead $w$ is chosen adaptively from the *same* evaluation data (e.g., the plug-in $\widehat{w}_{\mathrm{IVW}}$), the fixed-weight assumption is violated and the bound is no longer guaranteed. In such cases one can (i) revert to the union-bound inequalities of the main text, which hold for arbitrary (possibly data-dependent) $w$ via a triangle inequality; (ii) restore the fixed-$w$ condition through sample splitting or cross-fitting (Chernozhukov et al., 2018; Wager & Athey, 2018); or (iii) restrict $w$ to a finite grid $G \subset [0,1]$ and apply a union bound with $\delta$ adjusted by $|G|$ (Boucheron et al., 2013; Shalev-Shwartz & Ben-David, 2014).

## H  BENCHMARK DETAILS

From **MMLU-Pro** (Wang et al., 2024b), we select only the 10-option questions, yielding 9,795 examples. **MedQA-USMLE** (Jin et al., 2021) and **GPQA** (Rein et al., 2024) are both four-option multiple-choice benchmarks; for GPQA we adopt the extended version for broader coverage. For **MATH-MC**, following the candidate-generation protocol of Zhang et al. (2024), we construct five-option questions from the original **MATH** dataset (Hendrycks et al., 2021), discarding problems with fewer than four distractors, which yields 11,751 examples. All duplicates are removed to ensure consistency.

## I  ESTIMATOR CONFIGURATIONS

For completeness, we summarize the exact configurations of all estimators used in Table 1.

Ord  Ordinary-label estimator based on $n_{\mathrm{o}} = 300$ samples per dataset ($n_{\mathrm{o}} = 120$ for GPQA due to dataset size). Accuracy is computed directly from the proportion of correct ordinary labels.

Comp-$n_{\mathrm{o}}$  Complementary-label estimator with the same number of samples as the ordinary case, i.e. $n_{\mathrm{c}} = n_{\mathrm{o}}$ ($n_{\mathrm{c}} = 120$ for GPQA). This setting highlights the loss of efficiency when only complementary labels are available.

Comp-(K-1)  Complementary-label estimator with sample size fixed at $(K-1)$ times the ordinary case, i.e. $n_{\mathrm{c}} = (K-1)n_{\mathrm{o}}$. Following the protocol in Appendix C, each instance with oracle label $y$ is converted into $(K-1)$ complementary "No" labels $\{\bar{y} \in \mathcal{Y} \setminus \{y\}\}$, and the ordinary label is discarded; thus no ordinary labels are present in this setting.

Comp-Var  Complementary-label estimator with sample size $n_{\mathrm{c}}$ chosen such that its variance matches that of $n_{\mathrm{o}}$ ordinary labels, following Eq. (4). This represents the idealized regime where complementary labels are sufficiently abundant to offset their lower per-sample information.

IVW-0.5  A baseline inverse-variance weighted estimator with a fixed equal weight $w = 0.5$ between ordinary and complementary accuracies. This serves as a naïve combination strategy without using variance information.

IVW    Inverse-variance weighted estimator with closed-form plug-in weights (Eq. (6)), combining ordinary and complementary estimates optimally under the variance–information identities.

ML    Maximum-likelihood estimator obtained by solving the joint likelihood equation using both ordinary and complementary samples. Asymptotically equivalent to IVW but involving a nonlinear update (Appendix D).

Ord-Eval    Oracle reference: accuracy evaluated directly on the entire dataset using ordinary labels. This is shown only for comparison and is excluded from macro-averages.

**Notes on datasets.** † **Math** denotes **Math-MC** evaluated with `GPT-4.1-nano` instead of `GPT-5-nano`, due to reasoning-output constraints. ‡ **Math (CoT)** denotes the same **Math-MC** dataset evaluated under chain-of-thought prompting (applied only to Math; model=`GPT-4.1-nano`).

## J    Implementation details on ADAS and AFlow

To ensure a fair comparison, we constructed validation and test splits for each benchmark, following the numbers and ratios used in Hu et al. (2025). Specifically, we sampled 120 validation and 800 test examples for Math-MC and Medical Abstract, and 88 validation and 458 test examples for GPQA. For AFlow, we set the number of validation runs to 3 and used the same validation set as ADAS, rather than the separate validation subset reported in Zhang et al. (2025b).

For code and prompts, we largely follow the official implementations of Hu et al. (2025) and Zhang et al. (2025b), with two minor modifications for reproducibility in our setting.

**ADAS.** The set of experts was adapted to better align with our benchmarks (e.g., replacing domain-specific experts in LLM Debate with ones relevant to GPQA and Medical Abstract). All other prompt components remain unchanged. We uploaded the full text of the adapted prompts and scripts in the supplementary material.

**AFlow.** Since the original AFlow does not directly correspond to the multiple-choice format, we modified the answer extraction and normalization functions, and used the following prompt:

```
Answer the following multiple choice question.
{problem['Question']}
(A) {problem['A']}
(B) {problem['B']}
(C) {problem['C']}
(D) {problem['D']}

At the very end, output the final answer in the format:

### X

Where X is exactly one of A, B, C, or D.

Do not add anything after this line. The last line of your output must
only contain '### X'. If you write anything after the line '### X', your
answer will be marked as incorrect.
```

## K    Estimator Variance Analysis

Figure 4 reports ADAS and AFlow under the two estimators—raw complementary accuracy ($q$) and its linear transform (trans)—each selected at the iteration that maximizes validation accuracy. The results show clear regime dependence: on **GPQA** (the hardest benchmark), $q$ is more reliable (e.g., ADAS drops from 47.2 with $q$ to 42.4 with trans, while AFlow remains unchanged at 47.2). On **Math-MC**, ADAS still prefers $q$ (88.1 vs. 84.2), whereas AFlow shows only a slight gain with trans (86.8 vs. 86.5). On **Medical Abstract**, where complementary accuracy is relatively high, the transform becomes beneficial: ADAS improves from 61.6 to 67.8, and AFlow from 68.5 to 68.9.

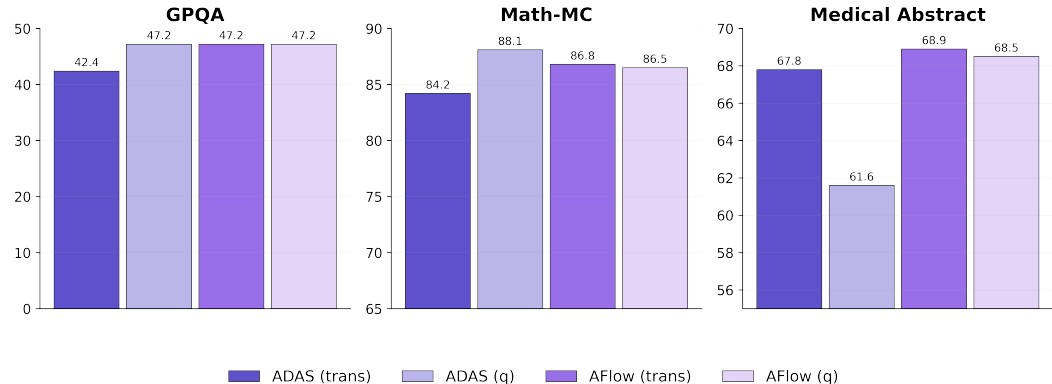

Figure 4: **Estimator sensitivity under weak signals.** Detailed comparison of raw complementary accuracy ($q$) and its linear transform (trans) across GPQA, Math-MC, and Medical Abstract. The figure illustrates how the two estimators behave differently depending on task difficulty, with variance amplification explaining when the transform helps or hurts.

This behavior is explained by variance amplification: according to Eq. (3), the variance of the transformed estimator scales by $(K-1)^2$ compared to $q$. When signals are weak (e.g., GPQA), variance amplification exacerbates noise and harms performance. In settings with stronger signals (Math-MC, Medical Abstract), the effect is less detrimental, and the transformed estimator occasionally yields modest gains. A systematic study of this trade-off is left for future work.

## L DEVIATION FROM THE ORD-EVAL REFERENCE

For completeness, Table 3 reports the absolute deviation ($\Delta$) between each estimator and the Ord-Eval oracle on each benchmark. Column-wise minima are shown in **bold**. These values show that mixture estimators (IVW and ML) reduce the gap to the Ord-Eval reference compared with both ordinary-only and complementary-only estimators across most benchmarks, indicating that they provide more robust estimates. Ord and Comp-$n_o$ can occasionally produce point estimates that are numerically closer to Ord-Eval (as in MedQA and GPQA). However, as discussed in Section 3.1, their standard errors are substantially larger, suggesting poorer reliability and limited reproducibility. This further supports the robustness of the mixture estimators.

Table 3: Absolute deviation ($\Delta$; in percentage points) from the Ord-Eval oracle. Column-wise minima are shown in **bold**.

| Estimator | MMLU-Pro | MedQA-USMLE | GPQA | MATH | MATH(CoT) | Avg $\Delta$ |
|---|---|---|---|---|---|---|
| Ord | 0.36 | 0.23 | 4.65 | 3.35 | 1.00 | 1.92 |
| Comp-$n_o$ | 0.97 | **0.01** | **0.35** | 4.23 | 3.45 | 1.80 |
| Comp-Var | 2.30 | 2.05 | 4.15 | 3.11 | 2.54 | 2.83 |
| IVW-0.5 | 0.08 | 1.05 | 5.76 | 0.77 | 0.33 | 1.60 |
| IVW | **0.00** | 0.80 | 5.62 | 0.66 | **0.03** | **1.42** |
| ML | 0.03 | 1.01 | 5.59 | **0.54** | 0.24 | 1.48 |

## M CASE STUDY: BEHAVIOR OF AGENTS TRAINED WITH PARTIAL FEEDBACK

In this section, we visualize and analyze the internal mechanisms of the agentic systems produced by ADAS and AFlow on the Medical Abstracts dataset. This case study illustrates how agentic workflows learned under partial feedback acquire coherent, structured decision processes. Our goal is to show that, even with weak supervision, these systems diverge meaningfully and improve from fixed workflows.

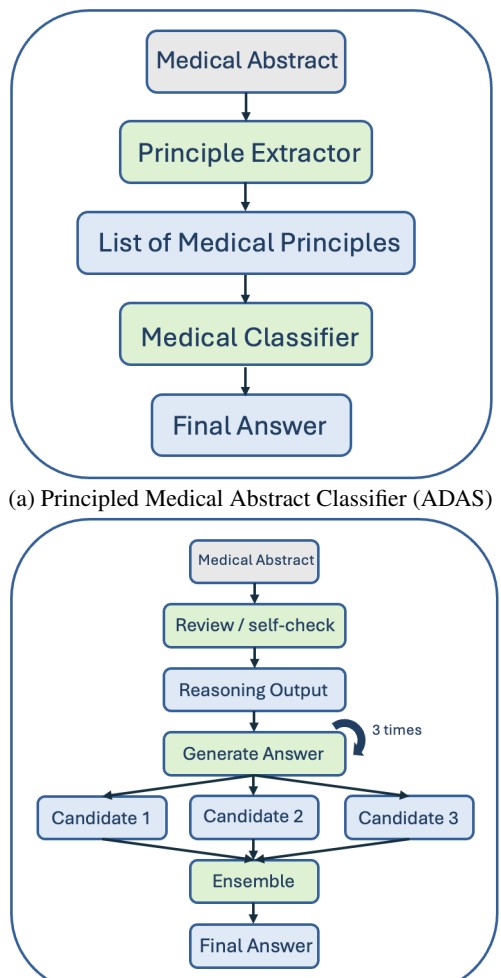

(a) Principled Medical Abstract Classifier (ADAS)

(b) Review-Generate–Ensemble Workflow (Aflow)

Figure 5: Workflows of the best ADAS and AFlow agents learned under partial-feedback training. Complementary label rewards lead ADAS to adopt a principle-extraction–guided classifier, while AFlow learns a review-generate–ensemble pipeline.

Figure 5 presents the best-performing agentic workflows discovered by ADAS and AFlow. The ADAS system, which we refer to as the Principled Medical Abstract Classifier, and the AFlow workflow, which name was chosen for clarity, both adopt multi-stage decision pipelines rather than directly producing a single-shot answer, as done by baseline methods.

In Figure 5(a), ADAS performs a consistent two-stage decomposition. Instead of mapping an abstract directly to a disease label, the agent first produces a numbered list of key medical principles or mechanistic factors extracted from the text. These principles are then passed to a separate classifier, which uses them as evidence for the final prediction. Figure 5(b) shows that AFlow adopts a multi-candidate reasoning pipeline. The agent first generates a structured review of the abstract using a task-specific prompt. Conditioned on this shared reasoning trace, it then produces several candidate answers. A final ensemble operator aggregates these candidates and selects the most consistent prediction.

Such modular, review-based, and multi-candidate workflows do not appear in established baselines, which rely on fixed or single-pass reasoning. The emergence of these structures demonstrates that complementary label supervision can shape the agent's decision process, not merely its accuracy. Partial feedback encourages the agent to construct interpretable intermediate representations and use them as evidence, even without full supervision. This confirms that complementary label training provides a behavior shaping signal that yields more structured and interpretable workflows.

## N    LARGE LANGUAGE MODELS USAGE STATEMENT

LLMs assisted us in formulating our ideas, working on extensions and analysis, and polishing the writing. All of the LLM outputs were subsequently verified, revised and modified by the authors. Any errors or inappropriate statements remain the responsibility of the authors.

