# OpenReview forum: "Towards Scalable Oversight via Partitioned Human Supervision"
_ICLR.cc/2026/Conference — ICLR 2026 Poster_

### Official Review · Reviewer_oJ9E · 2025-10-24

**Soundness:** 4
**Presentation:** 4
**Contribution:** 3
**Rating:** 6
**Confidence:** 2

**Summary:**

This paper addresses the growing challenge of evaluating and supervising AI systems that exceed human expert capabilities. The authors propose Partitioned Human Supervision, a scalable oversight framework in which multiple experts each provide complementary labels, indicating which candidate answers are incorrect, rather than full ground-truth annotations. The paper derives unbiased estimators for model accuracy using only these partial signals, and further introduces mixture estimators that combine complementary and ordinary labels. Experiments in financial and medical domains demonstrate that such supervision can reliably estimate model performance and even serve as a weak training signal when full supervision is infeasible.

**Strengths:**

- The paper tackles a timely and important problem: how to scale human oversight when AI systems operate beyond individual expertise. The framing of ``partitioned’’ expertise is both realistic and conceptually strong.
- The derivation of unbiased estimators for accuracy under complementary labeling is clear and mathematically grounded. The analysis of variance and the proposed IVW and MLE-based estimators are rigorous and interpretable.
- The approach can significantly reduce supervision cost by replacing full labeling with weaker but cheaper expert judgments. This has meaningful implications for high-expertise domains like medicine or law.
- The authors evaluate their method on domain-specific benchmarks, showing that it remains reliable even when each expert only rules out incorrect answers. The results support the method’s robustness and real-world applicability.

**Weaknesses:**

- The framework suits multi-choice or structured decision problems but may not extend naturally to open-ended or generative tasks where correctness is ill-defined.
- The method assumes experts can accurately identify incorrect options. In practice, experts may err or have partial uncertainty, which could bias the estimators. The paper could better analyze the effect of noisy or correlated human feedback.

**Questions:**

- The framework can involve experts who reliably provide negative feedback. How would it handle cases where experts are uncertain or only partially knowledgeable? Does the estimator remain valid under probabilistic or noisy complementary signals?
- Does partitioned supervision assume disjoint expert domains? If experts’ competencies overlap, how are conflicting or redundant complementary labels managed, and what effect might this have on estimator consistency?
- The method is tested on structured tasks. Could it extend to open-ended or generative settings by redefining complementary feedback (e.g., flagging logically inconsistent outputs)?

---

> ### Author Response · Authors · 2025-11-21
> **Initial Response Part 1**
>
> We thank the reviewer for the clear and constructive assessment. The appreciation of the partitioned-expertise formulation, the rigor of the estimator analysis, and the practical relevance for high-expertise domains is encouraging. We address the reviewer’s concerns point-by-point below.
>
> ---
>
> # Applicability to Open-Ended or Generative Settings
> Thank you for raising these broader questions about the motivation and scope. Many open-ended or highly complex problems can, in practice, be decomposed into structured decision steps, even if the final task appears generative or ill-defined. In real-world evaluation pipelines, models often interact with multiple subsystems, and it is common to diagnose errors by isolating intermediate choices or sub-decisions. Framing such checkpoints in a multiple-choice or hierarchical format provides a practical and interpretable way to localize uncertainty and leverage partial domain knowledge. This makes structured evaluations a useful building block for oversight, even in settings where the end-to-end task is open-ended. We agree, however, that this abstraction does not capture the full space of open-ended behavior, and we have added a limitations paragraph discussing when complementary labels remain informative, how structured formulations may fail to generalize, and what extensions would be required for more open-ended tasks.
>
>
> # Addressing Noisy Expert Feedback
> Thank you for highlighting this important point. The assumption of uniformly querying an index is indeed central to obtaining unbiased complementary labels. In our labeling protocol, this can be implemented by the system dispatching queries uniformly at random across indices, while experts only need to confirm whether the shown option is incompatible with the correct answer within their domain. Since the expert’s task is binary and local (i.e., to rule out a single option), the probability of introducing systematic bias is comparatively small.
>
> Nevertheless, we agree that this assumption should be stated more clearly. We will revise the setup in Section 2 to explicitly discuss how uniform querying is operationalized in practice. Additionally, we will add a paragraph in the discussion/limitations section outlining potential extensions that relax this assumption. Prior work on biased complementary labels [A], suggests promising directions for developing estimators that remain robust even under non-uniform querying.
>
> [A] Yu, X., Liu, T., Gong, M., & Tao, D. (2018). Learning with biased complementary labels. In Proceedings of the European conference on computer vision (ECCV) (pp. 68-83).
> [B] Wang, W., Ishida, T., Zhang, Y. J., Niu, G., & Sugiyama, M. (2023). Learning with complementary labels revisited: The selected-completely-at-random setting is more practical. arXiv preprint arXiv:2311.15502.

---

> > ### Author Response · Authors · 2025-11-21
> > **Initial Response Part 2**
> >
> > # Handling Uncertain or Partial Expertise
> > Thank you for the question. While our framework assumes that experts provide reliable complementary feedback under uniform querying, we agree that in practice experts may be uncertain or only partially knowledgeable, leading to probabilistic or noisy complementary signals. Our current estimator is designed for the unbiased setting, but prior work on noisy complementary labels [C] provides techniques for handling such uncertainty. Incorporating these ideas would allow the estimator to remain valid under probabilistic feedback and represents a natural extension of our framework. We have added a brief discussion of this point in the revised version.
> >
> > [C] Ishiguro, H., Ishida, T., & Sugiyama, M. (2022). Learning from noisy complementary labels with robust loss functions. IEICE TRANSACTIONS on Information and Systems, 105(2), 364-376.
> >
> >
> > # Managing Overlapping Expert Domains
> > Thank you for raising this point. Our current formulation assumes that each instance receives a single complementary label, which implicitly corresponds to querying one expert domain. In practice, overlapping expertise could lead to multiple or potentially conflicting complementary labels. Although this setting is not the focus of our paper, prior work on learning with multiple complementary labels [D] provides a principled way to handle such redundancy and maintain estimator consistency. Extending our framework to incorporate these ideas would be a natural direction for making the method more robust in practical multi-expert environments. We have added a brief discussion of this in the revised version.
> >
> > [D]: Feng, L., Kaneko, T., Han, B., Niu, G., An, B., & Sugiyama, M. (2020, November). Learning with multiple complementary labels. In International conference on machine learning (pp. 3072-3081). PMLR.
> >
> >
> > # Extending to Open-Ended or Generative Settings
> > Thank you for raising this question. This point is closely related to the broader issues we addressed above regarding open-ended or generative settings. As discussed, many seemingly open-ended tasks can be decomposed into structured intermediate decisions, which allows complementary feedback to be defined over these local checkpoints. In principle, redefining complementary feedback to flag properties such as logical inconsistency or domain-level violations is compatible with our framework, as these signals can be viewed as ruling out subsets of hypotheses in the same way that “not cardiology’’ rules out a domain. However, we agree that extending the formulation to fully open-ended settings—where such structure may not exist or is hard to extract—requires additional work. We have added a discussion in the limitations section outlining when such feedback remains informative and what further extensions would be needed for genuinely open-ended generative tasks.
> >
> > ---
> > We thank the reviewer for the constructive insights and for giving us the chance to clarify our ideas. We believe the revisions and explanations offered here directly tackle the identified issues and support a more positive overall view of the submission.

---

> > > ### Comment · Reviewer_oJ9E · 2025-11-25
> > >
> > > I appreciate the authors’ clear and thorough response. As my concerns have been sufficiently addressed, I am raising my score to accept.

---

> > > > ### Author Response · Authors · 2025-11-25
> > > >
> > > > We are grateful that our response was able to sufficiently address your concerns, and we sincerely appreciate your decision to raise the score. Thank you for your constructive comments and helpful feedback.

---

### Official Review · Reviewer_KfQX · 2025-10-28

**Soundness:** 3
**Presentation:** 3
**Contribution:** 3
**Rating:** 8
**Confidence:** 2

**Summary:**

This work aims to improve LLM evaluations in settings where we have limited access to expert annotations by using complementary labels (derived by asking experts to evaluate a model output as either confidently in or out of their expert class). These complementary labels are weaker than ordinary labels as they provide less information and have high variance as compared to ordinary labels, but this work formalizes the impact that these labels can have in evaluation by introducing 2 estimators for combining ordinary and complementary labels (IVW and ML), deriving associated Hoeffding/Bernstein bounds for the estimators, extensive empirical evaluation over traditional and more real-world datasets, and agentic training using the estimator as a fitness function. This work is answering a timely question of how do we leverage humans in LLM evaluations, especially in cases where human annotation is scarce, which is becoming more likely as LLM capabilities increase.

**Strengths:**

This work regarding how we can leverage human data in nontraditional ways via complementary labels is original and timely to the current dearth of adequate evaluation paradigms for LLMs. The paper leverages high-quality evaluation paradigms to support their claims, approaching the problem both empirically (across multiple benchmarks, tasks, and domains) and theoretically (unbiased estimator derivation, bounding the variance of the estimators, quantifying how many complementary labels equate to a singular ordinary label) and deriving strong and consistent results. The work is generally clear, although I discuss within the weakness/questions the parts that would benefit from additional detailing and motivation. The work also has potential for significance in the field of scalable oversight and evaluations, as more instances arise where obtaining explicit human feedback from a desired expert may be infeasible but alternative, weaker forms of feedback may be more scalable and readily available.

**Weaknesses:**

One of my biggest qualms with the work is the general motivation framework. “...future models will tackle problems whose solutions are too technical or too crossdisciplinary for any single human to verify comprehensively. When we cannot produce ground truth or prepare automated verifiers, how should we evaluate and train such systems?” In theory there is value to this question and motivation, but how do we ensure that “not cardiology” is an informative label? Specifically, in this toy example, why is cardiology feedback available but the true specialist is not? Additional discussion of why we may expect this paradigm in medicine and in what instances (Is this a diagnosis problem? If it is, would we ever truly have a multiple choice set where we cannot define the true label but we know the complementary label? Then how did we get the set of choices?) . As such, if we are entering a paradigm beyond human expert understanding, is multiple choice the best setting in which to be evaluating this set-up? And if not, how would this system generalize to more open-ended evaluation tasks? It would be helpful to discuss these limitations in the conclusion as potential future work.

**Questions:**

1. Please provide extended explanations for why MathMC results in Figure 3 are not as strong as on the other benchmarks. What is it about this benchmark that makes the complementary labels less effective?

2. In Table 1, it is not clear what the estimator methods are. I see that the methods are outlined in the appendix, but there needs to be some discussion/legend present in the main text as well. Further, the presentation of the table generally can be improved.  Can you highlight how much mixture helps beyond just regular usage of fewer ordinary labels? You also could just report the delta directly between the oracle instance and the different methods.

---

> ### Author Response · Authors · 2025-11-21
> **Initial Response**
>
> We appreciate the reviewer’s positive and detailed assessment. The comments on the originality of leveraging complementary labels, the combination of strong theory and broad empirical support, and the relevance to scalable oversight are very encouraging. We address the reviewer’s questions and concerns below.
>
> ---
>
> # Clarifying the Motivation and Medical Example
> Thank you for raising these broader questions about the motivation and scope. Our medical example is intended as an idealized abstraction of real diagnostic workflows: it captures the core decision structure rather than reproducing a full deployment scenario, where the actual options and flow may be substantially more complex. In such settings, there may in principle be many subspecialists (cardiology, neurology, oncology, etc.), but it is unrealistic, both logistically and economically, to query every possible expert about whether a case falls within their domain. Instead, clinicians often provide partial information such as “this is unlikely to be cardiology,” which we model as a complementary label. Our approach is precisely designed to learn from such partial feedback, without assuming access to all relevant specialists or ground-truth labels for every instance. The multiple-choice format in our toy example serves as a convenient abstraction for reasoning about how partial domain feedback constrains the hypothesis space. In practice, the candidate set often comes from standard diagnostic workflows (e.g., differential diagnosis lists), even when the precise ground truth is uncertain. This allows complementary feedback to be informative despite the absence of a definitive label. We agree that these scenarios require additional discussion, and we have added a limitations paragraph in the updated version addressing the applicability of complementary labels, the role of multiple-choice formulations, and possible extensions toward more open-ended evaluation tasks.
>
>
> # Interpreting Performance on MathMC
>
> Thank you for the question. We attribute the different behavior of the MathMC benchmark relative to the other datasets to two main reasons.
>
> First, MathMC is already a highly saturated task under CoT prompting, leaving limited headroom for further gains from modifying the reward signal. As also observed in prior work (e.g., AFlow on GSM8K), mathematical reasoning datasets tend to exhibit smaller improvements from reward-level adjustments compared to other domains.
>
> Second, as discussed in Appendix K (Estimator Variance Analysis), the variance of complementary-label estimators can interact with tasks whose option structure is highly fine-grained, which may make improvements less pronounced in MathMC relative to broader knowledge benchmarks.
> Our goal in this experiment is to demonstrate that partial feedback remains usable for agentic training, and these characteristics of MathMC help explain why the observed improvements are smaller than in other settings
>
> [A] Zhang, J., Xiang, J., Yu, Z., Teng, F., Chen, X., Chen, J., ... & Wu, C. (2024). Aflow: Automating agentic workflow generation. arXiv preprint arXiv:2410.10762.
>
>
> # Improving Table 1 Presentation
> Thank you for the helpful suggestions on the presentation of Table 1. We have added a brief discussion and legend in the main text to clarify the estimator methods. We also highlight the improvements from the mixture approach by reporting deltas relative to the oracle baseline, and we provide an additional comparison table in the appendix. These updates are included in the revised version.
>
> ---
>
> We are grateful for the reviewer’s detailed comments and for the opportunity to refine the presentation of our work. It is our hope that the responses and adjustments outlined above effectively address the concerns and improve the paper’s overall impression.

---

### Official Review · Reviewer_2oJj · 2025-10-31

**Soundness:** 4
**Presentation:** 2
**Contribution:** 3
**Rating:** 6
**Confidence:** 2

**Summary:**

This paper proposes a scalable oversight framework to evaluate and train advanced AI systems using "complementary labels," which are weak signals from experts identifying incorrect options rather than the correct ground truth. The authors derive estimators to measure accuracy from these weak signals and show how to combine them with scarce ordinary labels. Empirically, the results demonstrate this method can successfully evaluate LLMs without ground truth and can even be used to train an agentic AI system to perform better.

**Strengths:**

- This work studies a very relevant topic that is fundamental for the enhancement of current AI systems, especially for their training and evaluation w.r.t. humans.
- The paper provides mathematical details, motivation and theoretical analysis of its proposed framework. In this way, the contributions are concretely based on mathematical principles and characterization.
- The result demonstrating that weak human feedback can provide useful learning signal is very promising.

**Weaknesses:**

- Top-1 accuracy can be scarce or too limited to evaluate the performance of a foundation model.
- The presentation is a bit difficult to follows as it is very technical. However, I don't consider this a proper weakness.
- If I understand correctly, the number of complementary labels needed for a satisfactory training can be very large.
- I didn't understand if it is a typo, but several paragraph titles are coloured, which is quite uncommon.

**Questions:**

1) Can the authors comment about the limitations of their approach?
2) Assuming full ground truth is scarce, what is an estimation of the number of weak supervision required to get the same performances?
3) What the authors think is the reach of the different learning tasks that a LLMs can learn by using weak supervisions only, or weak supervisions + a few full ground truth supervisions?

---

> ### Author Response · Authors · 2025-11-21
> **Initial Response Part 1**
>
> We are grateful for the reviewer’s thoughtful evaluation. The recognition of the work’s relevance, its mathematical grounding, and the promise of learning from weak human feedback is appreciated, and we welcome the opportunity to address the raised points in detail.
>
> ---
>
> # Justification about Top-1 accuracy
> We understand this comment as raising three related concerns:
> (1) whether our method can handle top-k accuracy,
> (2) whether multiple-choice evaluation is too limited for assessing foundation models, and
> (3) whether it is enough to use top-1 accuracy to evaluate generated agentic systems.
>
> (1) Applicability to top-k accuracy.
> Our method naturally extends to top-k accuracy, as top-k correctness can be decomposed into k sequential applications of top-1 correctness. Therefore, the same complementary-label estimation framework applies without modification.
>
> (2) Is multiple-choice evaluation too limited?
> While frontier LLMs are indeed evaluated on coding or generative tasks, multiple-choice benchmarks remain core metrics across nearly all recent model releases. For example, both GPT-5 [A] and Gemini 2.5 Pro [B] continue to report performance on GPQA-Diamond, MMMU-Pro, and the multiple-choice subset of Humanity’s Last Exam. Thus, the multiple-choice format is still widely regarded as an essential and standardized component of LLM evaluation.
>
> (3) Evaluation on agentic systems.
> To address this concern, we additionally include a case study demonstrating that an agent trained with partial (complementary) feedback can learn effectively under our framework.
>
> [A] https://openai.com/index/introducing-gpt-5/
> [B] https://deepmind.google/models/gemini/pro/
>
>
> # Number of Complementary Labels Required:
> Thank you for raising this point. This issue is indeed analyzed in Section 2.1 (Estimator from Complementary Labels). In particular, Equation (5) provides a quantitative characterization of how many complementary labels are needed to match the information contained in ordinary labels:
>
> $n_c = \left(1 + \frac{K-1}{A}\right) n_o$,
>
> where $n_c$ is the number of complementary labels, $n_o$ is the number of ordinary labels, K is the number of answer options, and A is the true accuracy.
>
> As the reviewer suggests, complementary labels are individually less informative than ordinary labels, and thus more samples are typically required. However, we argue that the annotation cost per complementary label is substantially lower. This cost advantage is precisely why complementary labels can still be preferable in practice.
>
> To further support this claim, Section 3 reports experiments (Fig. 1) showing that Comp-Var, when using the number of complementary labels predicted by Equation (5), achieves performance comparable to the ordinary-label baseline.

---

> > ### Author Response · Authors · 2025-11-21
> > **Initial Response Part 2**
> >
> > # Scope of Learning Under Weak or Hybrid Supervision:
> > As we clarify in our other responses, we provide equations that explicitly connect performance under ordinary labels to that under complementary labels. Moreover, for weak supervision combined with a small number of fully supervised examples, Section 2.2 (Combining Ordinary and Complementary Labels) outlines how we estimate accuracy in this setting.
> > We also present extensive experiments in Section 3 (Fig. 1) demonstrating the effectiveness of our approach. In addition, Section 2.3 provides theoretical analysis, including bounds for the proposed estimators; please refer to Theorems 2 and 3 for details.
> >
> > Therefore, under learning scenarios where accuracy is the evaluation metric and a sufficient number of complementary labels is available, we believe that weak supervision—either alone or augmented with a small amount of full supervision—can achieve performance comparable to using ordinary labels. Prior work has also analyzed loss functions and bounds for learning solely from complementary labels, such as Ishida et al.  [C, D].
> >
> > [C] Ishida, T., Niu, G., Hu, W., & Sugiyama, M. (2017). Learning from complementary labels. Advances in Neural Information Processing Systems, 30.
> > [D] Ishida, T., Niu, G., Menon, A., & Sugiyama, M. (2019, May). Complementary-label learning for arbitrary losses and models. In International conference on machine learning (pp. 2971-2980). PMLR.
> >
> >
> > # Discussion of Method Limitations:
> > Thank you for the question. Our approach does have limitations, primarily regarding the scenarios in which complementary labels are available and applicable. In settings where obtaining even complementary labels is difficult, or where evaluation is highly open-ended and less structured, the framework may require additional adaptation. To make these points explicit, we have added a dedicated Limitations section in the updated version of the paper.
> >
> >
> > # Clarity of Presentation:
> > Thank you for the comment. We acknowledge that the technical nature of our method may make the presentation dense. In the camera-ready version, we will improve the exposition and provide additional intuitive explanations. We also plan to release a public blog post with accessible, non-technical summaries to make the approach easier to follow for a broader audience.
> >
> >
> > # Formatting Issue:
> > Thank you for pointing this out. The colored paragraph titles were a stylistic choice intended to improve readability and avoid overusing subsubsections. However, since this formatting is uncommon in ICLR submissions, we have updated the revision to follow the conventional style.
> >
> > ---
> >
> > We appreciate the reviewer’s thoughtful feedback and the chance to clarify our contributions. We hope the explanations and revisions above adequately address the concerns and help strengthen the overall assessment of the work.

---

> > > ### Comment · Reviewer_2oJj · 2025-11-25
> > >
> > > Thanks a lot for your clarifications. I'll raise my score to full acceptance.

---

> > > > ### Author Response · Authors · 2025-11-25
> > > >
> > > > We are pleased to hear that our rebuttal has successfully addressed your concerns, and we sincerely appreciate your decision to raise the score. Thank you very much for your constructive feedback and valuable time.

---

### Official Review · Reviewer_t5Fc · 2025-11-01

**Soundness:** 3
**Presentation:** 3
**Contribution:** 3
**Rating:** 6
**Confidence:** 4

**Summary:**

The paper studies how do we evaluate AI systems when direct verification is difficult (as the abilities of an AI may exceed that of any single human). The paper relies on the notion that ruling out an incorrect answer is easier than verifying that the answer is correct. It prposes a partitioned supervision protocol, where the human experts indicate when a particular answer is definitely incorrect rather than providing a correct answer (or actual verification). Using these complementary labels, with the additional assumption that these complementary labels are drawn uniformly at random from, the incorrect options, the paper proves that it is possible to construct an unbiased estimator for model top-1 accuracy using these labels, and variance estimators are also studied. Experimental results are convincing (but some questions remain).

**Strengths:**

1. I think the theoretical framing of complementary labels as sufficient to estimate model performance is clean and clearly motivated. I like that the paper focuses on the oversight bottleneck, which is timely.
2. The derivation of the estimator and the variance analysis are correct to me. The decomposition and reasoning about how the uniform query mechanism leads to unbiased estimation is neatly done. I also found the combination with ordinary labels to reduce variance to be intuitive.
3. The experimental evaluation is thorough within the context of multiple-choice settings. In that sense, the paper is rigorous in validating the correctness of the statistical estimator.

**Weaknesses:**

1. While I follow the theoretical contribution, I am less convinced about the real-world applicability for the kind of motivation the paper has. The experiments are exclusively on closed-form multiple-choice tasks. However, the motivating context in the paper is open-ended and superhuman domains. In that sense, there is a bit of a mismatch: the paper does not really demonstrate that the method scales to the intended setting.
2. I also think the paper assumes too much familiarity with the labeling protocol. The uniform selection of an index to query is crucial to guaranteeing unbiased complementary labels, but this assumption is not deeply discussed. In realistic scenarios, domain experts may not always be queried in such a uniform way.
3. The paper claims agentic training benefits using this estimator as a reward signal, but the empirical improvements there are small and relatively difficult to interpret. It would help to clarify what practical training improvements this enables and whether there are examples where the method meaningfully changes agent behavior.

Overall, I think the paper makes a meaningful and conceptually clean theoretical contribution. The estimator is sound and the controlled experiments support the theory. However, I think the paper could improve on motivating and validating how this method translates to settings beyond multiple-choice evaluation. I would benefit from stronger discussion on how the uniform querying assumption interacts with real-world annotation workflows.

**Questions:**

1. I had bit difficulty understanding the paper due to its notation. I believe, using $X$ and $Y$ for random variables and small letters $x, y$ for their realisations could help.

---

> ### Author Response · Authors · 2025-11-21
> **Initial Response Part 1**
>
> We appreciate the reviewer’s careful reading and the insightful remarks on the theoretical framing, the correctness of the estimator derivations, and the thoroughness of the empirical evaluation. The acknowledgment of the clarity of the complementary-label formulation and the soundness of the variance analysis is especially encouraging. We address the reviewer’s questions and concerns in detail below.
>
> ---
>
>
> # Applicability Beyond Closed-Form Tasks
>
> Thank you for the insightful comment. We agree that our current experiments focus on closed-form multiple-choice tasks, while the broader motivation of scalable oversight concerns more open-ended and superhuman domains. We address the concern as follows.
> First, despite rapid progress in open-ended evaluation, multiple-choice benchmarks remain a central and standardized tool for assessing frontier foundation models. Recent releases such as GPT-5 [A] and Gemini 2.5 Pro [B] continue to rely heavily on multiple-choice benchmarks (e.g., GPQA-Diamond, MMMU-Pro, subsets of Humanity’s Last Exam). This indicates that closed-form evaluation remains highly relevant within the current paradigm.
>
> Moreover, many open-ended or highly complex problems can in practice be reformulated into structured multiple-choice or hierarchical decision formats. For instance, real-world models, products, software evaluations often involve long pipelines that span numerous subsystems, making it difficult to directly pinpoint failure modes. In such cases, converting the diagnostic process into a sequence of localized, structured choices provides a practical and interpretable way to handle knowledge fragmentation across components. This further supports the relevance of closed-form evaluations as a building block for scalable oversight in open-ended domains.
>
> Second, we acknowledge that the complete solution of extending complementary-label estimation to open-ended settings is nontrivial and may require additional methodological innovations. To reflect this point clearly, we have added a dedicated Limitations section emphasizing that our current experiments do not yet demonstrate applicability to fully open-ended evaluations.
>
> Finally, we view this work as a first step toward establishing a scalable and data-efficient oversight mechanism. We believe the proposed framework offers a principled foundation on which future work can build to address more complex evaluation settings. To better reflect the current scope and avoid overstating claims, we have also revised the paper title to reduce potential mismatch between motivation and empirical setting.
>
> [A] https://openai.com/index/introducing-gpt-5/
> [B] https://deepmind.google/models/gemini/pro/
>
>
> # Practicality of the Uniform Index Selection
>
> Thank you for highlighting this important point. The assumption of uniformly querying an index is indeed central to obtaining unbiased complementary labels. In our labeling protocol, this can be implemented by the system dispatching queries uniformly at random across indices, while experts only need to confirm whether the shown option is incompatible with the correct answer within their domain. Since the expert’s task is binary and local (i.e., to rule out a single option), the probability of introducing systematic bias is comparatively small.
>
> Nevertheless, we agree that this assumption should be stated more clearly. We will revise the setup in Section 2 to explicitly discuss how uniform querying is operationalized in practice. Additionally, we will add a paragraph in the discussion/limitations section outlining potential extensions that relax this assumption. Prior work on biased complementary labels [C, D], suggests promising directions for developing estimators that remain robust even under non-uniform querying.
>
> [C] Yu, X., Liu, T., Gong, M., & Tao, D. (2018). Learning with biased complementary labels. In Proceedings of the European conference on computer vision (ECCV) (pp. 68-83).
> [D] Wang, W., Ishida, T., Zhang, Y. J., Niu, G., & Sugiyama, M. (2023). Learning with complementary labels revisited: The selected-completely-at-random setting is more practical. arXiv preprint arXiv:2311.15502.

---

> > ### Author Response · Authors · 2025-11-21
> > **Initial Response Part 2**
> >
> > # Interpreting Agentic Training Improvements
> > Thank you for the question. Our goal in the agentic setting is to provide that partial (complementary) feedback can serve as a usable reward signal. Since our estimator only replaces the reward component of the training pipeline, the scale of empirical improvement is naturally bounded by the underlying agent architecture and the original optimization dynamics.
> > That said, we agree that interpreting the behavioral impact is important. To provide clearer insight, we have added a case study illustrating how the agent’s action selection changes when trained with complementary-label rewards.
> >
> > Importantly, the improvements are not uniform across environments. Tasks where the baseline is not saturated, such as GPQA and Medical Abstracts, exhibit clearer gains, whereas datasets like MathMC, where the baseline model already performs at a high level, naturally show smaller deltas. This variation reflects differences in how much room there is for reward-signal improvements to influence training, rather than limitations of the estimator itself.
> >
> > To further contextualize the relevance of our setup, recent surveys on self-improving agent systems [E] highlight that a wide range of agentic pipelines (such as those implemented in our ADAS and AFlow experiments) rely heavily on derived or indirect supervision signals rather than fully explicit rewards. Our estimator provides exactly this type of partial-feedback supervision, making it directly applicable in agentic settings where complete labels are unavailable or costly. In this sense, the practical value of our method is less about achieving maximal performance gains, and more about enabling stable learning under realistic supervision constraints.
> >
> > Overall, the purpose of the agentic experiments is not to claim state-of-the-art performance, but to show that complementary-label rewards enable learning in scenarios where only partial feedback is available—thereby expanding the practical design space for self-improving systems and providing a drop-in reward mechanism compatible with existing training pipelines.
> >
> > [E]  Gao, H. A., Geng, J., Hua, W., Hu, M., Juan, X., Liu, H., ... & Wang, M. (2025). A survey of self-evolving agents: On path to artificial super intelligence. arXiv preprint arXiv:2507.21046.
> >
> >
> > # Extending Beyond Multiple-Choice Settings
> > Thank you for the feedback. At the same time, there are real-world scenarios where a multiple-choice formulation is both natural and practically relevant for scalable oversight. For example, in our medical domain setting, differential diagnosis can be cast as a structured multiple-choice problem: even experienced clinicians may misdiagnose due to domain specialization, making the task challenging and, in some cases, effectively superhuman for narrow experts. We will expand the discussion on the applicability beyond multiple-choice settings and provide additional clarification on how the uniform querying assumption relates to practical annotation workflows. These points will be addressed in the revised version.
> >
> >
> > # Improving Notational Consistency
> > Thank you for the suggestion. We have revised the notation in the updated version to better distinguish random variables from their realizations.
> >
> >
> > ---
> > Thank you for the insightful remarks and for allowing us to further elaborate on our contributions. We trust that the clarifications and updates provided here resolve the raised questions and contribute to a more favorable evaluation of the paper.

---

### Author Response · Authors · 2025-12-02
**Discussion Summary and Final Author Remark**

We thank the Area Chair for their time and all reviewers for their constructive, insightful feedback. We sincerely appreciate how the reviewers recognized our work as theoretically grounded, empirically validated, and offering a novel approach to a frontier problem. We believe their input has significantly improved the quality of our submission. Since the discussion window has now closed, below is a concise summary of how we responded, the changes we made, and how we addressed the main concerns raised during the review period.

---

**Practical validity of the uniform-distribution assumption**
We emphasized that the uniform query-distribution assumption is realistically achievable in deployed systems, since the system’s design disperses queries. This “dispersion by design” helps ensure the uniform assumption holds in practice, making it more credible and justifiable.

**Handling noisy, biased, or overlapping labels**
We acknowledged the reviewers’ concerns about label noise, bias, and overlap. In response, we pointed to existing prior work on biased complementary labels, noisy complementary labels, and multiple complementary-label settings, all of which are compatible with our framework. We argued that integrating such methods can strengthen the robustness of our system against these issues in the future.

**Added a practical case study: agentic training improvements**
To demonstrate real-world relevance and feasibility, we included a new case study showing how our method improves agent behavior in an agentic training setup. This addition helps illustrate that our approach can discover meaningful structure and yield practical benefits. (See details in Appendix M)

**Broader applicability beyond closed-form tasks**
We clarified that even open-ended problems, such as those involving multiple sections, can be reformulated as binary “check” phases or cast into a multiple-choice style, a format still used as a main evaluation metric by leading frontier models. We also explained that our framework has sufficient potential to extend to generative tasks, and added a discussion/limitations paragraph (See details in Section 5) to clearly state the motivation, scope, and boundaries of our current work.

**Formatting, notation, and presentation improvements**
We responded to reviewers’ remarks regarding format and notation: we refined the notation, reorganized sections for clarity, and ensured compliance with ICLR style guidelines. This should make the paper more readable and aligned with conference expectations.

---

We are grateful to the reviewers who responded to our rebuttal and expressed satisfaction with the revisions. We also acknowledge that, due to the discussion freeze, other reviewers who may have wished to reply no longer have the opportunity to do so. Thank you very much for your time and thoughtful consideration. We hope that the clarifications and revisions provided above will be taken into account during the final evaluation.

Thank you once more for the careful consideration.

---

### Meta-Review · Area_Chair_znGv · 2026-01-02

**Summary:**

This paper proposes a scalable oversight framework, Partitioned Human Supervision, to address the growing challenge of evaluating and supervising AI systems that may exceed human expert capabilities. All reviewers recommend acceptance. The main concerns in the initial reviews focus on the paper's motivation, the extent to which the results generalize to broader settings, and the assumption of uniformly querying an index. In the revision, the authors address several of these points and, where limitations remain, they acknowledge them explicitly and add further discussion. Overall, given the paper's valuable theoretical contributions and solid empirical support, there is no substantive opposition to acceptance.

**Reviewer Concerns:**

Reviewer 2oJj and Reviewer oJ9E respond to the rebuttal and indicate that their concerns have been addressed.

Reviewers t5Fc and KfQX raise questions about the paper's motivation and the extent to which the results generalize beyond closed-form tasks. In the revision, the authors strengthen the motivation, clarify the scope, and explicitly discuss possible extensions to settings beyond closed-form tasks, while acknowledging the current limitation.

Reviewers t5Fc and oJ9E also question whether the assumption of uniformly querying an index is overly strong or unrealistic. The authors' response seems to have convinced Reviewer oJ9E, and I likewise consider this concern to be satisfactorily addressed.

**Reviewer Scores:**

Both Reviewer 2oJj and Reviewer oJ9E engage during the discussion period and state that they plan to raise their scores to 8.

Reviewer KfQX assign a relatively high initial score of 8, and it is likely that the reviewer will maintain this positive evaluation.

It is less clear whether Reviewer t5Fc will increase their score. While the authors made meaningful efforts to address the raised concerns, some issues reflect inherent limitations of the current work (which the authors explicitly acknowledge). As a result, this reviewer may choose to keep the score unchanged, and it is unlikely that Reviewer t5Fc will lower it.

---

### Decision · Program_Chairs · 2026-01-26

Accept (Poster)